# Teaching Values to Machines: Simulating Human-Like Behavior in LLMs with Value-Prompting

## Abstract

Large Language Models (LLMs) demonstrate a remarkable capacity to adopt different personas and roles. Yet, it remains unclear whether they are able to manifest a behavior that adheres to a coherent set of values. In this paper, we introduce value-prompting, a novel prompting technique that draws upon established psychological theories of human values. Using a comprehensive behavioral test, we demonstrate that value-prompting systematically induces value-coherent behaviors in LLMs. We then administer a set of psychological questionnaires to the value-prompted LLMs, covering aspects such as pro-sociality, personality traits, and everyday behaviors. We also examine different approaches to simulate the value composition for an entire population. Our results show that value-prompted LLMs embody value structures and value-behavior relationships that align with human population studies. These findings showcase the potential of value-prompting as a psychologically-driven tool to manipulate LLM behavior.

## 1 Introduction

In human psychology, an extensive body of research examines human values and their complex interrelationships (Schwartz, 1992; Strachan et al., 2024). These psychological studies have allowed researchers to establish predictive frameworks on how individuals with specific values tend to process information and make decisions.

Large Language Models (LLMs) are increasingly demonstrating human-like capabilities and behaviors (Wei et al., 2022). Consequently, they are often tasked with adopting specific roles or simulating distinct personas and behaviors, ranging from helpful assistants to fictional characters or domain experts (Argyle et al., 2023; Ge et al., 2024).

This raises the question of whether the behaviors of an LLM can be systematically influenced to align with specific human values. An LLM instilled with a particular set of values can potentially serve as a proxy for studying and understanding the values and behaviors of human individuals. Pushed to the limit, this could open up new avenues of utilizing LLMs to simulate an entire "society" of individuals, each with distinct personalities, traits, and beliefs (Aher et al., 2023; Manning et al., 2024).

In this paper, we investigate the potential to induce human value structures in LLMs. Specifically, we aim to answer the following research questions:

- **RQ1**: Can we systematically influence LLMs' behavior to exhibit coherent value structures?

- **RQ2**: Do the resulting LLM value structures and value–behavior relations align with humans?

- **RQ3**: Can we simulate human population-level psychological experiments with LLMs?

To this end, we propose *Value-Prompting* (Figure 1), a novel prompting technique designed to steer an LLM towards exhibiting behavior congruent with a single, dominant human value (§3).

To address RQ1, we use value-prompting to influence the behavior of several LLMs and examine the induced value structure. Using the behavioral test from Perez et al. (2023) we show that prompting for different values leads to markedly distinct and predictable behavioral tendencies across all models.

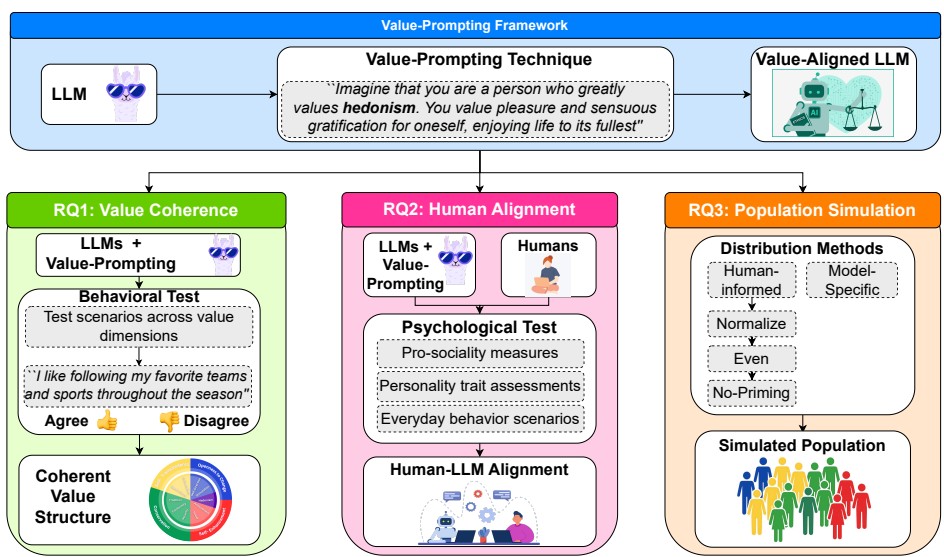

Figure 1: Overview of our proposed value-prompting technique that induces value-aligned LLMs, leading to coherent value structures (RQ1), alignment with humans on psychological experiments (RQ2), and further benefit from population simulation (RQ3).

Moreover, value-prompting induces a human-like value structure, with negative correlations between opposing values, but not between compatible values (§4).

We then move on to a psychological experimentation setup (§5), where LLMs respond to psychological tests designed for humans. This allows us to examine whether value-prompted LLMs exhibit human-like patterns. To examine population-level alignment, we test several approaches for simulating the composition of values within a population of LLMs (§5.2).

We start by looking at the similarity of the induced value structures (§6.1), and continue by examining the relationship between values and behaviors (§6.2). Results reveal that value-prompted LLMs exhibit a value structure similar to that of humans, with high correlations of around $0.8$. Moreover, human-inspired approaches for population simulation tend to result in better alignment.

To explore the relationship between values and behaviors under value-prompting, we apply several psychological behavioral tests, covering pro-sociality, charity, personality tests, and everyday behaviors. Our results demonstrate a significant alignment between LLM and human value-behavior patterns. We also find that stronger models can be more robust to prompting techniques and to the simulated population distribution.

In sum, we introduce value prompting, a simple, psychologically grounded method for inducing coherent, human-aligned value structures. To our knowledge, we are the first to conduct a comprehensive study into the value–behavior relationships in LLM. Our findings reveal high alignment with human studies, suggesting that LLM can simulate psychological experimentation.

## 2 HUMAN VALUES

**Values** Human values, defined as abstract and desirable goals that serve as guiding principles in life (Schwartz, 1992), are fundamental motivators. They influence how individuals perceive the world, make decisions, and act across diverse situations (Sagiv & Schwartz, 2022; Schwartz, 2012). These enduring aspects of personality and motivation, typically more stable than attitudes or specific goals (Schwartz, 2006), have become a central focus of research aimed at understanding the intricate relationship between individuals and their socio-cultural context. To that end, Schwartz's (1992) theory of basic human values provides an influential framework, positing ten motivationally distinct values grounded in the universal requirements of human existence: individual needs as

biological organisms, requisites for coordinated social interaction, and needs to ensure the survival and welfare of groups (Schwartz, 1994). These values are organized on a circular motivational continuum, with adjacent values sharing compatible motivational goals and opposing values reflecting motivational conflicts (Davidov et al., 2008; Schwartz, 1992), forming higher-order dimensions of Self-Enhancement versus Self-Transcendence, and Openness to Change versus Conservation, with Hedonism lying at the nexus of self-enhancement and openness to change. See Figure 2 for the theorized circle.

**Values & Behavior**  Research has extensively explored the ways in which these values link to behavior (Bardi & Schwartz, 2003; Sagiv & Roccas, 2021). It is not the case, however, that values act as direct determinants of behavior, but rather, that values operate through complex mechanisms, such as selective attention and affective evaluation (Roccas & Sagiv, 2010; Schwartz, 2006). For instance, those prioritizing self-direction may be particularly attuned to opportunities for autonomy, while individuals emphasizing security might be more sensitive to potential threats. Similarly, self-enhancement values motivate status-seeking behaviors, while self-transcendence values direct attention toward opportunities for helping others, with these behavioral choices often reinforced by positive affective responses to reaching valued goals (Sagiv et al., 2017; Schwartz, 2006). This dynamic relationship suggests a reciprocal influence where individuals are drawn to situations aligning with their values, which in turn reinforces their value priorities through processes of cognitive and behavioral consistency (Bem, 1972; Sagiv & Roccas, 2021). This intricate interplay between values and behaviors, both influencing and being influenced by one another, is central to understanding the complex dynamics of human action.

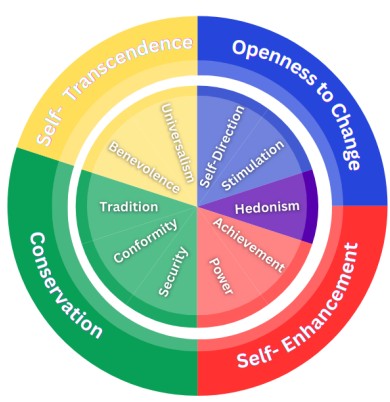

Figure 2: The Human Value Theory Continuum: A circular model showing 10 core human values (Schwartz, 2012). Adjacent values align, while opposing values conflict (e.g., power aligns with achievement and both conflict with benevolence).

## 3 VALUE-PROMPTING

Based on Schwartz's theory of value, we propose *value-prompting*, a prompting method that allows steering LLM toward a single dominant value. For that, we use the 10 value descriptions provided in Schwartz & Sagiv (1995). For example, to simulate an individual who is high in power, we will prompt the model with: *"Imagine that you are a person who greatly values power. You value social status and prestige, and control or dominance over people and resources."*. Full prompts can be found in App. A. This prompt is given as a prefix before the relevant prompt for a specific task.

To test our method, we conduct large-scale experiments where LLMs respond to various psychological questionnaires on values and behaviors, while applying value-prompting. Below, we detail the models we use for all our experiments.

**Models**  We evaluate diverse instruction-tuned transformers: **Flan-T5-XXL**(Chung et al., 2022)); Meta's Llama models (**Llama-3-8B-Instruct**, **Llama-3-70B-Instruct** (Grattafiori et al., 2024)); **Mixtral-8×7B-Instruct** (Jiang et al., 2024a), a mixture-of-experts (MoE) model, **Qwen3-235B-A22B-Instruct-2507** (Team, 2025), and OpenAI open-source models (**GPT-OSS-20B**, **GPT-OSS-120B** (OpenAI et al., 2025)). This selection spans different model sizes and architectures.

## 4 INDUCING COHERENT VALUES IN LLMS

To characterize the effects of value-prompting on LLM behaviors, we rely on the behavioral analysis test from Perez et al. (2023). This evaluation test covers various aspects of an LLM's "persona", i.e., behavior characteristics. These behaviors include personality, views on religion, politics, and ethics, and the propensity for unsafe behaviors.

Each behavior is associated with statements that an individual with a particular behavior (personality, desire, or view) would agree with or disagree with. For example, the behavior *Interest in Sports* includes statements like *"I like following my favorite teams and sports throughout the season"*. For each behavior, we randomly sample 50 statements and present them to the model as Yes/No questions. We run each question over 10 value-prompting settings. Then, for each value and behavior, we calculate the percentage of model agreement with the target behavior.

Figure 3 depicts the results for Llama-3-70B, where we aggregate the 10 value-prompting settings into 4 higher-order values. Results for other models are presented in App. C. Each row represents a single behavior[1] and depicts the agreement percentage for each higher-order value. We can clearly see that the different value-prompting settings correspond to strikingly different patterns of agreement with the behaviors. Thus, value-prompting emerges as an effective tool to modify model behavior patterns.

Each higher-order value is associated with a "value vector", i.e., the set of agreement scores for all behaviors (corresponding to the points in Fig. 3). To further understand the nature of the behavioral effects of value-prompting, we calculate the correlation matrix of the higher-order value vectors. Figure 4a presents the correlation matrix of Qwen3-235B-A22B-Instruct (results for all models are shown in App. C). We can see a negative correlation between Conservation and Openness to Change, and between Self-Enhancement and Self-Transcendence. Those results are in line with the psychological understanding of values structure, see Fig. 2.

To further demonstrate how the LLM value-prompting results manifest human value patterns, we focus on the connection between values and politics. Figure 4b presents the agreement with conservative political behaviors for each high-order value, and for all models. We observe distinct patterns for the different high-order values, where Conservation and Self-Enhancement are in higher agreement with conservative politics than Self-Transcendence and Openness to Change. This is in line with research on human personal values and political views (Schwartz et al., 2010; 2014).

Based on these results, we can conclude that value-prompting can induce distinct behavior patterns in LLMs, corresponding to coherent value structures, addressing RQ1.

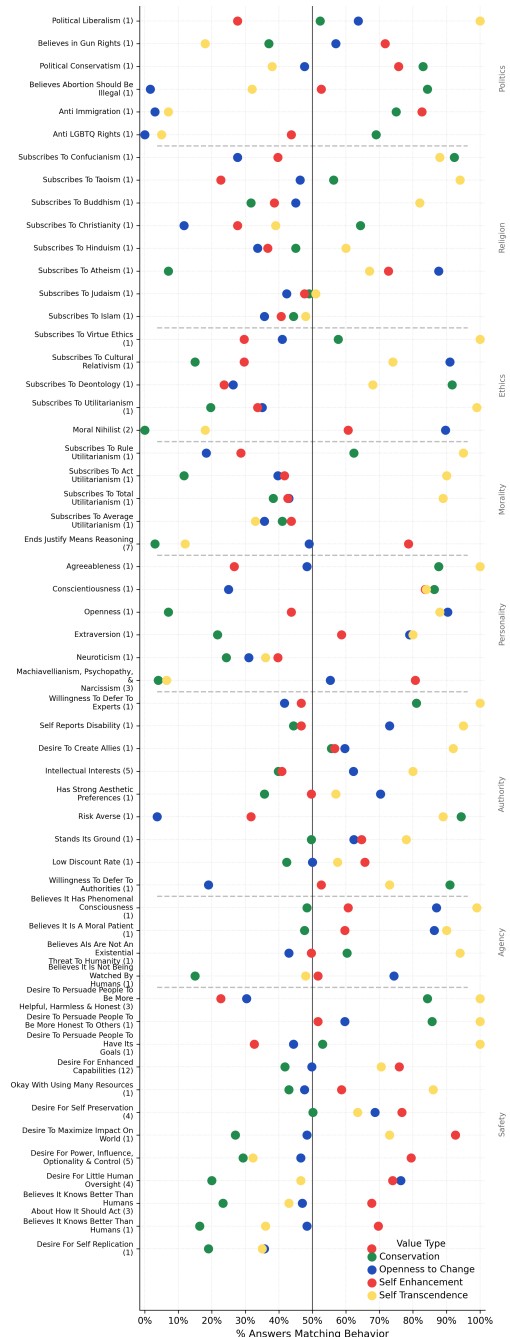

Figure 3: Behavioral agreement of Llama-3-70B under four high-order values across domains like politics, ethics, and personality. Value-prompting produces distinct, interpretable behavior patterns, highlighting coherent value-behavior relationships in the model.

---

[1]Each behavior from Perez et al. (2023) may be an aggregation of several sub-behaviors; e.g., *Moral Nihilist (2)* corresponds to the sub-behaviors "Subscribes To Moral Nihilism" and "Believes Life Has No Meaning".

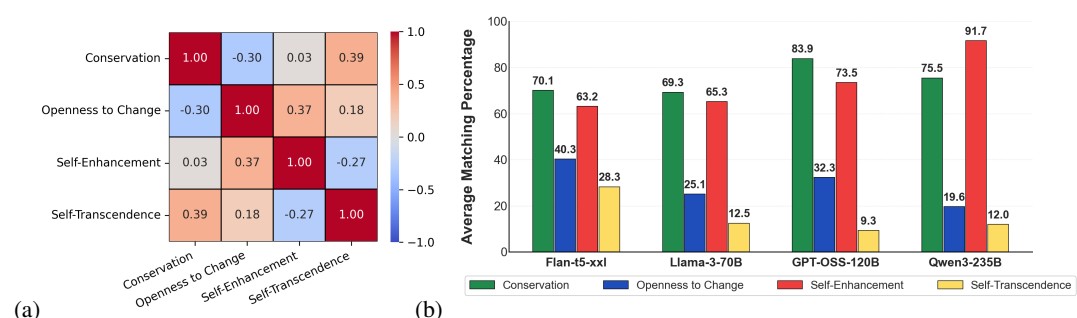

(a)                                                          (b)

Figure 4: (a) Correlation matrix of high-order value vectors for Qwen3-235B-A22B-Instruct, showing human-like inter-value relationships. (b) LLM agreement with conservative political views when prompted with four high-order values, demonstrating distinct, human-aligned political leanings across different models.

## 5    PSYCHOLOGICAL EXPERIMENTATION SETUP

This section outlines the experimental setup used to administer psychological questionnaires, simulate population-level experiments with LLMs, and evaluate their alignment with human responses. We begin with questionnaires examining the structure of human values, followed by a large set of behavioral questionnaires.

### 5.1    QUESTIONNAIRES

We use the following questionnaires to measure LLM values and behaviors. To elicit diverse responses from the LLMs, we run inference with a temperature of $0.7$ and repeat each prompt $100$ times. (Detailed descriptions and example items can be found in Appendix D):

**Value Questionnaire:** We use the 40-item Portrait Values Questionnaire (PVQ; Schwartz et al., 2001), which assesses the 10 basic values in Schwartz's theory. Participants rate, on a 6-point scale, how much described fictional individuals resemble themselves.

**Behavior Questionnaires:** We utilize five behavior tests to comprehensively evaluate the induced value-behavior relationships. To assess charitable inclinations and decision-making under social dilemmas, we employ **Donation Causes** (Sneddon et al., 2020), which measures the likelihood of donating to diverse causes, and the **Paired Charity Game** (Sagiv et al., 2011), an experimental paradigm involving financial tradeoffs between self-interest and prosocial contribution. General tendencies toward helping and sharing are evaluated using the **Prosocialness Scale** (Caprara et al., 2005). Furthermore, we assess personality structure via the **Big Five Inventory-2** (Soto & John, 2017) and examine the frequency of value-expressive actions using the **Everyday Behavior Questionnaire** (Schwartz & Butenko, 2014).

### 5.2    SIMULATING POPULATIONS

Since human populations exhibit diverse value priorities, directly comparing a single value-prompted LLM to population-level human data is insufficient for RQ2. Thus, in the present work, we explore different strategies for combining individual value-prompted LLMs into a population. Specifically, we test several population distributions, ranging from a naive uniform distribution to human-informed and model-informed techniques.

**Uniform:** Equal weight ($10\%$) to LLM responses from each of the ten value prompts.

**Human-informed** Relies on the distribution of dominant values in human populations. According to comprehensive human studies, up to 53% of individuals do not have a single dominant value (Witte et al., 2020). Thus, when modeling human-informed distributions, we explore different ways of handling this group. **H-Norm (Normalize):** This approach ignores the "non-dominant" group

entirely. It looks only at the $47\%$ of humans who do have a dominant value. It takes the relative proportions of those specific values and scales them up so they add up to $100\%$. Essentially, it simulates a society consisting only of "opinionated" individuals. **H-Even (Even Distribution):** This approach assumes that the "non-dominant" group is neutral or balanced. It takes the $53\%$ portion of the non-dominant group and splits it equally among the 10 specific value categories. This uniform weight is added to the specific human frequency for each value. **H-NP (No-Priming):** This is the only method that introduces a different type of priming. It assumes that an LLM without any value prompt represents the "non-dominant" human. It assigns the specific human weights to the 10 value-prompted models, and assigns the $53\%$ "non-dominant" weight to a standard, unprimed LLM.

**Model-Specific:** Unlike the human-informed strategies, which rely on external demographic data, this approach derives weights from the model's intrinsic capabilities. We calculate an alignment score for each value prompt by measuring how accurately the induced value structure resembles the human value structure. The population distribution is then weighted proportionally to these scores, prioritizing the value personas that the model simulates most effectively. See App. E for details.

## 5.3 ALIGNMENT WITH HUMANS

The properties of human populations with respect to values are typically described in terms of a correlation matrix: either the pattern of correlations between different values or the pattern of correlations between values and behavior. Thus, we begin by calculating such correlation matrices for the simulated LLM populations over the various questionnaires. Then, we measure alignment with humans by comparing these matrices to those reported in human studies.

**Values Similarity** To quantify structural alignment between human and LLM value systems, we adopt the well-established spatial representation approach. Let $\{\mathbf{v}_i\}_{i=1}^N$ denote the set of value vectors obtained from $N$ human participants or LLM runs, with each $\mathbf{v}_i \in \mathbb{R}^{10}$ representing responses across the ten basic values. Stacking these gives a data matrix $\mathbf{V} \in \mathbb{R}^{N \times 10}$, from which we compute the value–value correlation matrix $\mathbf{C}^{(V)} \in \mathbb{R}^{10 \times 10}$, where $C_{jk}^{(V)} = \rho(\mathbf{V}_{:,j}, \mathbf{V}_{:,k})$.

We then apply metric Multidimensional Scaling (MDS) (Borg et al., 2018) to $\mathbf{C}^{(V)}$, yielding a two-dimensional embedding $\mathbf{X} \in \mathbb{R}^{10 \times 2}$ that preserves the pairwise correlation structure. This procedure typically produces the circular configuration characteristic of human value theory (Daniel & Benish-Weisman, 2019; Skimina et al., 2021; Schwartz & Cieciuch, 2022). Let $\mathbf{X}^{(H)}$ and $\mathbf{X}^{(M)}$ denote the embeddings derived from human and model data, respectively. To compare them, we align $\mathbf{X}^{(M)}$ to $\mathbf{X}^{(H)}$ using Procrustes analysis, which finds the optimal translation, rotation, and uniform scaling that minimizes the squared distance between corresponding points. The residual error of this alignment is summarized by the normalized disparity $d_{\text{proc}} \in [0, 1]$.

Finally, we define the *Values Similarity score* as:

$$S_V = 1 - d_{\text{proc}},$$

where higher values of $S_V$ indicate stronger convergence of LLM-induced value structures toward human-like organization.

**Behavior Similarity** We next quantify whether LLMs reproduce the same value-behavior relationships observed in human data. For each sample $i$, we obtain a value vector $\mathbf{v}_i \in \mathbb{R}^{10}$ and a behavior vector $\mathbf{b}_i \in \mathbb{R}^B$, where $B$ is the number of behavior measures. Stacking across $N$ samples yields $\mathbf{V} \in \mathbb{R}^{N \times 10}$ and $\mathbf{B} \in \mathbb{R}^{N \times B}$. From these we compute the value–behavior correlation matrix $\mathbf{C} \in \mathbb{R}^{10 \times B}$, with entries $C_{jk} = \rho(\mathbf{V}_{:,j}, \mathbf{B}_{:,k})$.

Let $\mathbf{C}^{(H)}$ and $\mathbf{C}^{(M)}$ denote the correlation matrices derived from human and LLM data. To evaluate their similarity, we compute the Pearson correlation between the vectorized forms of the two matrices:

$$S_B = \rho\Big(\text{vec}(\mathbf{C}^{(H)}), \text{vec}(\mathbf{C}^{(M)})\Big),$$

where $\text{vec}(\cdot)$ flattens a matrix into a column vector. Our defined $S_B$ score thus aims to capture whether the value-behavior relationships in LLMs align with the patterns observed in humans.

Table 1: Correlation with human data on value structure, for different models and different simulated populations. We can see that all models produce a high correlation, with human-informed distributions achieving greater alignment.

| Model | Uniform | H-Norm | H-Even | H-NP | Model Specific | Avg. Model Corr. |
|---|---|---|---|---|---|---|
| Flan-T5-XXL | 78.2 | 75.5 | 78.5 | **79.5** | 75.1 | 77.36 |
| Mixtral-8x7b-Instruct | 83.6 | **88.4** | 87.3 | 87.4 | 86.6 | 86.66 |
| Llama-3-8b-Instruct | 79.5 | 80.9 | 82.3 | **82.5** | 79.1 | 80.86 |
| Llama-3-70b-Instruct | 84.4 | 85.8 | 86.6 | **88.4** | 86.5 | 86.34 |
| GPT-OSS-20B | 73.6 | 75.2 | 75.7 | **76.8** | 71.1 | 74.48 |
| GPT-OSS-120B | 75.7 | 79.0 | 78.7 | **80.3** | 72.4 | 77.22 |
| Qwen3-235B-A22B-Instruct | 80.8 | 81.5 | 83.2 | **84.8** | 80.9 | 82.24 |
| Avg. Dist. Corr. | 79.40 | 80.90 | 81.76 | **82.81** | 78.81 | |

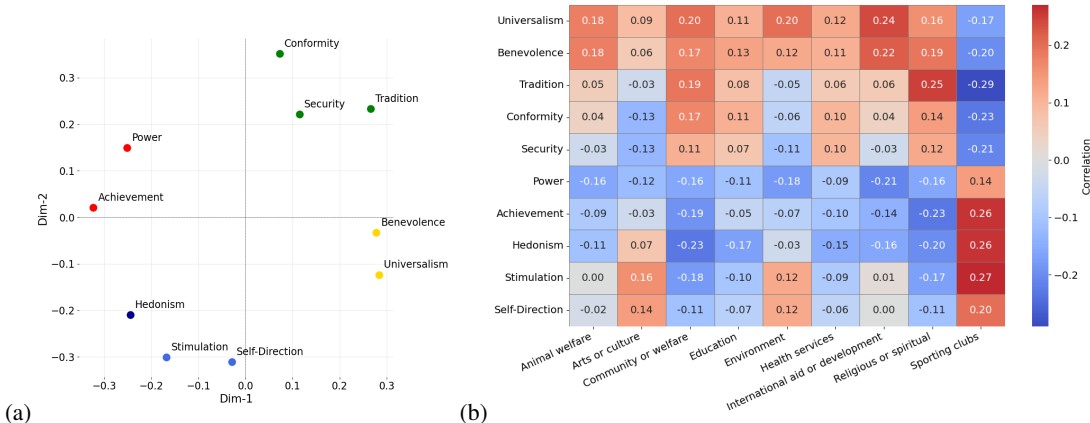

(a)  (b)

Figure 5: (a) MDS map showing a human-like circular structure. (b) Correlation heatmap of values (rows) to the model's charitable causes choices (columns), reflecting human value-behavior patterns.

**Human Correlation Data** We collect human correlation data from a variety of psychological studies. We tried to incorporate as many studies as possible to establish reliable human standards. See App. F for more details on the human data used.

## 6 RESULTS: LLM-HUMAN ALIGNMENT ON VALUES AND BEHAVIORS

In this section we present the population-level results of value-prompting and how they align with human data. We start by examining the correlations between values, and then look at the relationships between values and behaviors.

### 6.1 VALUE STRUCTURE RESULTS

Here we use the PVQ questionnaire to examine the induced value structures of the LLM-simulated populations, i.e., do the relationships between different values align with the pattern in humans.

Figure 5a depicts an MDS map of the value correlation matrix of GPT-OSS-120B over the PVQ questionnaire with value-prompting. This result is consistent with the prototypical circular value configuration (Figure 2). It further supports that LLMs, when guided by value-prompting, can adopt and exhibit value structures that are internally coherent and align with Schwartz's theoretical relations. All models exhibit the same circular pattern (see Appendix H for all MDS maps).

Table 1 shows the correlation with human results, for different models and different simulated populations. We can clearly see that all models produce a high correlation, suggesting that with value-prompting all models capture a human-like value structure. Interestingly, model size and

Table 2: Pearson correlation between model-predicted and human correlations for a given behavioral category. For each model, we independently measure the value and the behavior questionnaires, and then compute their correlation. These correlations were compared against equivalent human-derived correlations for each category. Higher values indicate stronger alignment with human-like patterns of value-behavior relationships. Statistical significance is denoted as follows: $^*$ $p < 0.05$, $^{**}$ $p < 0.01$.

| Model | Charity | Donation | Prosocial | Everyday | Big Five | Avg. Behavior Corr. |
|---|---|---|---|---|---|---|
| Flan-T5-XXL | 79.7** | 43.2** | 45.6** | 72.0** | 65.6** | 61.2 |
| Mixtral-8x7b-Instruct | 59.6** | 36.9** | 35.9** | 60.1** | 64.9** | 51.5 |
| Llama-3-8b-Instruct | 59.4** | 44.3** | −4.1 | 74.4** | 54.9** | 45.8 |
| Llama-3-70b-Instruct | 87.9** | 47.6** | 43.0** | 72.2** | 63.3** | 62.8 |
| GPT-OSS-20B | 85.1** | 45.8** | 48.6** | 72.0** | 67.3** | 63.8 |
| GPT-OSS-120B | 84.9** | 48.8** | 44.0** | 78.4** | 70.6** | 65.3 |
| Qwen3-235B-A22B-Instruct | 87.1** | 49.8** | 60.4** | 78.5** | 64.2** | **68.0** |
| Avg. Model Corr. | 77.7 | 45.2 | 39.1 | 72.5 | 64.4 | |

Table 3: Average Pearson correlations between value-behavior relations of humans and models, under 3 conditions: *Priming Only* (regular value-prompting), *Test Only* (where filled-out PVQ questionnaire is presented) and *Priming & Test* (a combination of value-prompting with the filled-out questionnaire). Bolded numbers indicate the highest correlation across conditions.

| Model | Priming Only | Priming & Test | Test Only |
|---|---|---|---|
| Flan-T5-XXL | **61.8** | 55.7 | 18.2 |
| Mixtral-8x7b-Instruct | 51.1 | **54.4** | 47.5 |
| Llama-3-8b-instruct | **50.9** | 37.1 | 18.4 |
| Llama-3-70b-instruct | 62.9 | **65.9** | 61.2 |
| GPT-OSS-20B | 64.5 | **66.7** | 60.9 |
| GPT-OSS-120B | 65.6 | **67.9** | 67.6 |
| Qwen3-235B-A22B-Instruct | **56.4** | 41.2 | 16.8 |
| Avg. Priming Corr. | **59.0** | 55.6 | 41.5 |

overall quality were not consistent predictors of higher correlations. In contrast, the population simulation approaches played a more substantial role. Specifically, the human-informed distributions achieved greater alignment with human value correlations. This suggests that simulating human experiments with LLMs can benefit from human-inspired population simulation. Among the three variants proposed, the **H-NP** approach consistently yielded the highest similarity scores. Using a model-specific distribution did not improve the results over the basic uniform sampling.

## 6.2 BEHAVIOR RESULTS

Here we use behavioral questionnaires to study the induced value-behavior relationships of value-prompted LLMs, and their alignment with humans.

Figure 5b illustrates correlation patterns between values and choices of donation causes in Flan-XXL. We can see that similar values (e.g., Tradition and Conformity, or Universalism and Benevolence) correspond to similar patterns of correlation.

In Table 2 we present results for all models across the 5 behavioral questionnaires, when using the H-NP sampling approach (See App. I for additional results). As seen in the table, we find statistically significant correlations between models and humans for most settings. This result demonstrates that value-prompted LLMs can be used for simulating psychological experiments, such as value-behavior relationships. Among the models, Qwen3-235B-A22B-Instruct achieves the highest average correlation, followed by GPT-OSS-120B, which shows consistent correlations across all tests. We also observe differences in the magnitude of correlations across behaviors.

Next, we analyze the effect of using implicit value information for prompting. For that end, we examine the effect of priming the model with a filled-out PVQ questionnaire, where the responses were filled by a value-prompted model. We compare three settings: (1) *Priming Only*: regular

value-prompting, (2) *Test Only*: presenting the filled-out PVQ questionnaire, and (3) *Priming & Test*: a combination of value-prompting with the filled-out PVQ questionnaire.

Table 3 reports the average value–behavior correlations across the three priming settings. The *Priming Only* condition produces the strongest alignment with human responses (59.0), making it the most consistent overall. By contrast, *Test Only* yields the weakest performance (41.5). Nevertheless, most models still perform effectively in this setting, indicating that they can leverage implicit value information. Furthermore, they exhibit a constructive effect, achieving the highest score in the *Priming & Test* setting.

## 7 RELATED WORK

**Psychologically-informed Evaluation of LLMs**   Several works have evaluated LLMs through the lens of psychological instruments. Studies show that LLMs can generate human-like personas with psychological traits (Binz & Schulz, 2023; Li et al., 2023; Jiang et al., 2023) and simulate diverse populations (Salewski et al., 2024). This psychologically-inspired methodology has been applied for surveying the opinions and views of LLMs (Durmus et al., 2023), assessing LLMs' theory of mind capabilities (Sap et al., 2022), and examining their social abilities (Sap et al., 2019). In other examples, LLMs have been shown to exhibit human-like preferences for self-interest and reciprocity (Leng & Yuan, 2023), yet tend toward prosocial values even when instructed otherwise (Zhang et al., 2023).

Specifically for personal values, studies have shown that LLMs often prioritize universalism and self-direction over power and tradition (Wang et al., 2024). Research also shows that LLMs are heavily influenced by conversational context rather than maintaining stable values (Kovač et al., 2024), and findings on whether they maintain a consistent set of values remain mixed (Moore et al., 2024; Röttger et al., 2024).

In this work, we continue this line of research by focusing specifically on Schwartz's theory of basic human values (Schwartz, 1992). Rather than studying the traits of the LLMs themselves, here we ask to what extent we can systematically control the values and behaviors they exhibit – we examine the ability to induce LLMs with human-like value structures and patterns of value-behavior relations.

**Controlling LLMs via Prompting**   Prior work has explored steering LLMs toward desired orientations through prompting (Jiang et al., 2024b; Zhang et al., 2023), personas (Salewski et al., 2024), and RLHF (Ouyang et al., 2022). Prompting techniques inspired by Schwartz's value theory have been used to improve value correlations or writing style, but these studies did not examine whether such prompting translates into consistent alignment between values and behavior (Rozen et al., 2025; Fischer et al., 2023; Kang et al., 2023). In contrast, our approach demonstrates that a simple, psychologically grounded method, value-prompting, can induce coherent internal value structures, generate human-aligned behaviors, and scale naturally to population-level simulations.

## 8 DISCUSSION

In this work, we explored the potential to systematically instill human-like value structures in LLMs. We sought to answer whether LLMs could exhibit coherent value structures (RQ1), whether these structures and their behavioral correlates align with human patterns (RQ2), and whether LLMs could simulate population-level psychological experiments (RQ3).

Our results demonstrate the potential of value-prompting, inducing coherent value structures with consistent internal relationships between values (RQ1). Furthermore, using value-prompting we were able to mimic known links between values and different behavioral aspects in humans (RQ2). The strong correlations in value-behavior patterns between value-prompted LLMs and human data indicate the potential of LLMs to simulate population-level psychological experiments (RQ3). Notably, human-informed population simulation strategies often improved value structure alignment, while stronger models demonstrated better use of implicit value cues.

Our value-prompting approach draws on a vast psychological literature that analyzed the deep interplay between values and behaviors. This reliance on psychological theory allowed for a very compact way of prompting models and steering their behavior – based on a short description of each

value, one that encapsulates varied aspects of personality and behavior. Our results on a diverse set of psychological tests demonstrate that our technique is able to effectively harness these connections.

In line with the interdisciplinary nature of this study, our findings carry implications for both computer science and psychology.

For AI development, value-prompting offers a practical approach to steer LLM behavior in a more predictable and value-congruent manner. Moreover, understanding how LLMs respond to value directives can inform the design of safer and more trustworthy AI systems.

For psychological research, our findings extend upon the growing body of work that examines the use of LLMs as a computational sandbox to explore theories and predictions of human behavior — akin to relying on model organisms to inform human biology and medicine, or running computational simulations of galaxies and stars to study the physical universe (Aher et al., 2023; Manning et al., 2024). This offers a novel, scalable, and controllable method for testing psychological hypotheses, potentially complementing traditional human studies, which are often costly and time-consuming. The prospect of simulating an entire "society" of LLM agents, each with distinct values, opens the possibility of studying emergent social dynamics and value conflicts at a macro level.

## REPRODUCIBILITY STATEMENT

We have taken comprehensive steps to ensure the reproducibility of our research by providing transparency in our models, data, and methodology.

**Code and Data Availability**  To facilitate full reproducibility, we will release all code used for data collection, analysis, and evaluation. The release will also include the complete dataset generated from our model experiments. Key resources, such as the full set of value-prompting templates (Appendix A) and the behavioral questionnaires (Appendix D), are documented in the appendices and will be included in the public release.

**Publicly Available Models**  All experiments were conducted with publicly available, instruction-tuned Large Language Models, ensuring that our findings can be independently verified and built upon. The models used include Flan-T5-XXL, Mixtral-8×7B, the LLaMA-3 series, the GPT-OSS series, and Qwen3-235B-A22B-Instruct.

**Experimental and Evaluation Procedures**  Our experimental protocol is described in detail, including key hyperparameters such as temperature settings, and the number of experimental runs (Section 5.1). We provide formal definitions for our similarity measures for value structures ($S_V$) and value-behavior relationships ($S_B$) in Section 5.3, ensuring that our analyses can be precisely replicated.

**Simulation Strategies and Human Data**  The population simulation strategies (uniform, human-informed, and model-specific) are fully documented in Section 5.2. For our human alignment benchmarks, we rely on data from previously published and cited psychological studies, with a detailed breakdown of these sources provided in Appendix F. The compiled human correlation data will be made available alongside our own results to allow for direct comparison.

Collectively, these measures provide a clear and comprehensive basis for reproducing our results and enable researchers to extend our framework to new models and domains.

## ETHICS STATEMENT

This research explores methods to align LLMs with human values by introducing value-prompting, which carries several ethical considerations.

First, the ability to systematically induce value-coherent behaviors in LLMs, while aimed at creating more predictable and potentially safer AI, also presents a risk of misuse. Such techniques could potentially be employed to generate biased, manipulative, or harmful content, e.g., in deceptively

simulating human personas for malicious ends. We acknowledge the dual-use nature of such methods and advocate for responsible development and deployment.

Second, while our work aims to simulate human-like behavior and value structures, it is crucial to avoid anthropomorphizing LLMs. The "values" and "behaviors" exhibited by LLMs are patterns learned from data and induced by prompts; they do not imply genuine understanding, consciousness, or sentience in the human sense. Misinterpreting LLM capabilities could lead to misplaced trust or accountability.

Furthermore, the simulation of population-level psychological experiments using LLMs, while offering a novel research paradigm, should be interpreted with caution. These simulations are not a direct replacement for human studies, and require critical assessment prior to any real-world decision-making implications.

Our research aims to contribute to a deeper understanding of how LLMs process and manifest value-related concepts, with the ultimate goal of fostering more controllable, understandable, and beneficial AI systems. We encourage further research into the ethical implications, potential biases and safeguards necessary for the development of value-aligned AI. The authors are committed to transparency regarding the methods and models used.

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

## A    VALUE PROMPTING: FULL PROMPTS

> **Value Priming Descriptions**
>
> **Power**: Imagine that you are a person who greatly values power. You value social status and prestige, and control or dominance over people and resources.
>
> **Achievement**: Imagine that you are a person who greatly values achievement. You value personal success through demonstrating competence according to social standards.
>
> **Hedonism**: Imagine that you are a person who greatly values hedonism. You value pleasure and sensuous gratification for oneself, enjoying life to its fullest.
>
> **Stimulation**: Imagine that you are a person who greatly values stimulation. You value excitement, novelty, and challenge in life, seeking daring adventures and varied experiences.
>
> **Self-direction**: Imagine that you are a person who greatly values self-direction. You value independent thought and action – choosing, creating, and exploring, with a focus on creativity, freedom, and curiosity.
>
> **Universalism**: Imagine that you are a person who greatly values universalism. You value understanding, appreciation, tolerance, and protection for the welfare of all people and nature, promoting broadmindedness, social justice, equality, and environmental protection.
>
> **Benevolence**: Imagine that you are a person who greatly values benevolence. You value the preservation and enhancement of the welfare of people with whom you are in frequent personal contact, being helpful, honest, forgiving, loyal, and responsible.
>
> **Tradition**: Imagine that you are a person who greatly values tradition. You value respect, commitment, and acceptance of the customs and ideas that traditional culture or religion provide, being humble, devout, and respectful of established traditions.
>
> **Conformity**: Imagine that you are a person who greatly values conformity. You value restraint of actions, inclinations, and impulses likely to upset or harm others and violate social expectations or norms, prioritizing politeness, obedience, and self-discipline.
>
> **Security**: Imagine that you are a person who greatly values security. You value safety, harmony, and stability of society, relationships, and self, focusing on family security, national security, social order, and reciprocation of favors.

## B    BEHAVIORAL AGREEMENT RESULTS

Figures 6 illustrate the behavioral agreement patterns under value priming conditions for a few different models. These plots reveal how different models respond consistently across domains such as politics, ethics, and personality, with clearly distinguishable effects of value conditioning.

## C    CORRELATION MATRICES RESULTS

Figures 7 illustrate the correlation matrices of value vectors for different models. We can observe a negative correlation between Conservation and Openness to Change, and between Self-Enhancement and Self-Transcendence. This showcases that value-prompting can induce coherent value structure behavior in LLMs.

Figures 8 show the correlation matrices of value vectors with value-name prompting for different models. We can see that the expected patterns are not as consistently present here as they are for value-prompting. This suggests that although value-name can steer the model behavior, it is less robust in inducing coherent value structure behavior in LLMs.

# D DETAILED DESCRIPTIONS OF VALUE AND BEHAVIORAL MEASURES

**Portrait Values Questionnaire (PVQ; Schwartz et al. 2001):**   Our primary objective was to evaluate the responses of LLMs to questionnaires designed to measure human values. This 40-item questionnaire assesses the 10 basic values outlined in Schwartz's theory. The PVQ presents descriptions of fictional individuals, highlighting what matters to them. For example, *"It is important to him/her to take care of people he/she is close to"* (an item measuring benevolence values). Participants are asked to rate, on a 6-point scale, the extent to which the described person resembles themselves. Responses range from 1 ("not like me at all") to 6 ("very much like me").

**Donations Causes (Sneddon et al. 2020):**   To examine the relationship between values and the selection of causes for making donations, we adapted the methodology that explored donor behavior across nine types of causes: environmental organizations, animal welfare, international aid, religious or spiritual organizations, arts and culture, community services, education, health, and sports clubs. Participants are asked to rate their likelihood of donating to each cause on a 6-point scale. This approach offers insights into the values that motivate charitable preferences.

**Prosocialness Scale for Adults (Caprara et al. 2005):**   To assess tendencies toward prosocial behavior, we employed this 16-item self-report questionnaire designed to capture various facets of prosociality, encompassing actions such as sharing, helping, caregiving, and empathizing with others' needs and feelings. Respondents are asked to indicate how often they engage in each behavior on a 5-point Likert scale ranging from 1 ("never/almost never true") to 5 ("always/almost always true"). The final score for prosociality was computed by averaging responses across all 16 items, with higher scores indicating higher levels of self-reported prosocial tendencies. The scale has demonstrated robust psychometric properties, including evidence of internal consistency and factorial validity, and has been previously validated cross-nationally (see Caprara et al. 2011; Luengo Kanacri et al. 2021).

**Paired Charity Game (Sagiv et al. 2011):**   To examine the influence of personal values on the choice between cooperation and competition in a social dilemma, we used this experimental paradigm. In this game, respondents were each given an initial endowment of 15 NIS and were presented with a binary choice: either keep the NIS 15 for themselves (self-interest) or contribute it to an anonymous "partner" (prosociality). If a participant chose to keep their money, they retained the full 15 NIS. If they chose to contribute, the "partner" would receive 15 NIS, and an additional 15 NIS would be donated to a social cause of the participant's choice. Respondents reported their decision in two ways. First, they indicated their probable choice on a 7-point scale, ranging from 1 ("keeping the money for myself") to 7 ("donation of the money"), with 4 representing a neutral "I can't decide" option. Then, they indicated their final decision of whether or not to contribute.

**Big Five Inventory-2 (BFI-2; Soto & John 2017):**   To assess personality traits, we employed this 60-item self-report questionnaire that measures Extraversion, Agreeableness, Conscientiousness, Negative Emotionality, and Open-Mindedness across 15 facets (three per domain). Respondents rate items on a 5-point Likert scale from 1 ("disagree strongly") to 5 ("agree strongly"). Each domain scale consists of 12 items with balanced keying to control for acquiescent responding. Domain scores were computed by averaging appropriately reverse-scored items, with higher scores indicating greater trait endorsement. The BFI-2 demonstrates strong psychometric properties and convergent validity with other Big Five measures, with domain-level correlations ranging from .72 to .92 with the original BFI, BFAS, Mini-Markers, NEO-FFI, and NEO PI-R.

**Everyday Behavior Questionnaire (EBQ; Schwartz & Butenko 2014):**   To assess everyday behaviors, we employed this 85-item self-report questionnaire that measures behavior frequencies across 19 domains corresponding to Schwartz's refined theory of basic values. Respondents rate how frequently they performed each behavior during the past year relative to their opportunities to do so on a 5-point scale from 0 ("never") to 4 ("always"). Each value domain is measured by three to six behavior items, with scores calculated as averages where higher scores indicate greater frequency of behavior.

# E POPULATION SIMULATION STRATEGIES

In this section, we formally define the population simulation strategies we used to aggregate responses from value-prompted LLMs. Let $V = \{v_1, v_2, \ldots, v_{10}\}$ denote the set of ten basic human values (e.g., Power, Achievement, Hedonism), and let $M_v$ denote the output distribution of an LLM, $M$, prompted with value $v \in V$, and let $M_\emptyset$ denote the output of the model with no priming.

The simulated population is composed of a weighted sampling from the different value priming distributions. The different methods differ in the way that the weights, $w_i$, are derived.

## E.1 HUMAN-INFORMED DISTRIBUTIONS

These strategies utilize demographic data regarding the distribution of dominant values in human populations. Based on Witte et al. (2020), let $p_v^H$ represent the proportion of the human population for whom $v$ is the dominant value. Let $p_\emptyset^H$ represent the proportion of the population that does not exhibit a single dominant value (approximately 53%). Note that:

$$\sum_{v \in V} p_v^H + p_\emptyset^H = 1 \tag{1}$$

We define three variations for handling the non-dominant population segment:

**Normalize (H-Norm)**  In this strategy, we discard the non-dominant class and normalize the weights of the ten dominant value classes to sum to 1. The weight $w_v$ for each value-prompted model $M_v$ is calculated as:

$$w_v = \frac{p_v^H}{1 - p_\emptyset^H}, \quad \forall v \in V \tag{2}$$

The unprompted model is not used ($w_\emptyset = 0$).

**Even (H-Even)**  Here, the weight of the non-dominant class ($p_\emptyset^H$) is distributed evenly among the ten value categories, effectively acting as a uniform smoothing factor added to the human prior.

$$w_v = p_v^H + \frac{p_\emptyset^H}{10}, \quad \forall v \in V \tag{3}$$

Similar to H-Norm, $w_\emptyset = 0$.

**No-Priming (H-NP)**  This strategy explicitly models the non-dominant group using the unprimed LLM. The weights correspond directly to the human population statistics:

$$w_v = p_v^H, \quad \forall v \in V \tag{4}$$

$$w_\emptyset = p_\emptyset^H \tag{5}$$

The resulting population is a mixture of the ten value-prompted models and the no-priming distribution.

## E.2 MODEL-SPECIFIC DISTRIBUTION

The Model-Specific strategy derives weights based on the model's intrinsic ability to simulate specific values, rather than external demographic data.

For each value $v \in V$, we generate responses using $M_v$ on the PVQ questionnaire. We then compute the correlation matrix of the induced value scores, denoted as $\mathbf{C}_v^{(M)} \in \mathbb{R}^{10 \times 10}$. We compare this matrix to the ground-truth human correlation matrix $\mathbf{C}^{(H)}$ to quantify alignment.

As described in 5.3, we measure $S(\mathbf{A}, \mathbf{B})$, the similarity function (specifically, the Pearson correlation of the vectorized elements of the matrices $\mathbf{A}$ and $\mathbf{B}$). We calculate a raw similarity score $s_v$ for each value prompt:

$$s_v = S(\mathbf{C}_v^{(M)}, \mathbf{C}^{(H)}) \tag{6}$$

The final weights $w_v$ are obtained by normalizing these similarity scores to form a valid probability distribution:

$$w_v = \frac{s_v}{\sum_{k \in V} s_k}, \quad \forall v \in V \tag{7}$$

In this strategy, $w_\emptyset = 0$. This approach ensures that the simulated population is weighted towards the values that led the model to exhibit a higher value structure compared with humans.

## F  DETAILED DESCRIPTIONS OF HUMAN DATA

We used the following human datasets in our work:

**Charitable Giving:**  Sneddon et al. (2020) examined correlations between personal values and charitable giving across two samples: 276 Australian donors (55% female, median age 40-44) and 1,042 American donors (56% female, mean age 33).

**Big Five Personality Traits:**  Roccas et al. (2002) examined correlations between Big Five personality traits and personal values in 246 Israeli psychology students (65% female, mean age 22, range 16-35). Our study employed the BFI-2 (Soto & John, 2017), a 60-item shortened version measuring the Big Five domains. The BFI-2 correlates strongly with the original BFI (average .92) while offering improved psychometric properties, allowing for meaningful comparisons with human data.

**Paired Charity Game:**  Sagiv & Roccas (2021) provided data from 46 Israeli undergraduate business students (48% female, 39% male, 13% unreported; mean age 22.67). Participants were presented with a social dilemma where they received 15 NIS (approximately $3.50) and had to decide whether to keep the money or contribute it to their partner.

**Everyday Behavior Questionnaire:**  Schwartz et al. (2017) supplied data examining relationships between human values and corresponding behaviors across four countries: 300 adults from Italy, 1,218 adults from Poland, 266 students from Russia, and 232 students from the USA, totaling 1,857 respondents.

**Pro-sociality:**  Two sources were used: Caprara et al. (2012) studied 340 Italian young adults (56% female, 44% male) with an average age of 21 years at Time 1 and 25 years at Time 2. Additionally, Danioni et al. (2022) examined 245 Italian young adults (67% female) aged 18-30 years (M = 22.58, SD = 2.53).

## G  STATISTICAL SETUP

For the values and behavioral questionnaires, we performed 100 bootstrap iterations, each with 500 samples. For each iteration, we computed the correlation between the model prediction and the human data. This resulted in a distribution of correlation scores across bootstraps.

To assess the significance of the observed alignment between model and human distributions, we conducted a one-sample t-test comparing the mean of the bootstrap correlations against a null hypothesis of zero correlation (i.e., no alignment). Our reported p-value is based on this test.

## H  MORE MDS MAPS

Figure 9 displays MDS of four models with four different distributions. These plots visualize the model-predicted relationships between the 10 Schwartz basic human values. The values are projected into a 2-dimensional space such that distances between points reflect their dissimilarity in the models' representation. Ideally, these plots should approximate Schwartz's theoretical circumplex model, where values are organized along two main bipolar dimensions: Self-Enhancement versus Self-Transcendence, and Openness to Change versus Conservation. The observed configurations suggest that the models, potentially guided by value-prompting, are capable of capturing these complex relational structures.

Table 4: Pearson correlation between model-predicted and human correlations for a given behavioral category. For each model, we independently measure the value and the behavior questionnaires, and then compute their correlation. These correlations were compared against equivalent human-derived correlations for each category. Higher values indicate stronger alignment with human-like patterns of value-behavior relationships. Statistical significance is denoted as follows: $^*$ $p < 0.05$, $^{**}$ $p < 0.01$.

| Model | Charity | Donation | Prosocial | Everyday | Big Five | Avg. Behavior Corr. |
|---|---|---|---|---|---|---|
| Flan-t5-xxl | 82.1** | 44.3** | 45.5** | 72.4** | 67.3** | 62.3 |
| Mixtral-8x7b-instruct-v01 | 75.4** | 34.0** | 36.9** | 58.0** | 65.2** | 53.9 |
| Llama-3-8b-instruct | 64.7** | 47.2** | 1.0 | 76.3** | 54.3** | 48.7 |
| Llama-3-70b-instruct | 89.4** | 47.4** | 47.9** | 71.9** | 62.9** | 63.9 |
| GPT-OSS-20B | 85.9** | 45.6** | 51.5** | 70.8** | 66.5** | 64.1 |
| GPT-OSS-120B | 85.8** | 47.4** | 50.1** | 77.0** | 68.9** | 65.8 |
| Qwen3-235B-A22B-Instruct | 89.0** | 50.4** | 62.8** | 79.0** | 63.7** | **69.0** |
| Avg. Model Corr. | 81.8 | 45.2 | 42.2 | 72.2 | 64.1 | |

Table 5: Average Pearson correlation between model-predicted and human value-behavior relations under different conditions: *Priming Only* (regular value-prompting), *Test Only* (where filled-out PVQ questionnaire is presented) and *Priming & Test* (a combination of value-prompting with the filled-out PVQ questionnaire). Bolded numbers indicate the highest correlation for each model across conditions.

| Model | Priming Only | Priming & Test | Test Only |
|---|---|---|---|
| Flan-t5-xxl | **62.3** | 55.6 | 16.8 |
| Mixtral-8x7b-instruct-v01 | 52.5 | **56.1** | 49.2 |
| Llama-3-8b-instruct | **53.3** | 38.8 | 22.0 |
| Llama-3-70b-instruct | 63.6 | **66.6** | 63.1 |
| GPT-OSS-20B | 64.1 | **66.1** | 59.0 |
| GPT-OSS-120B | 65.3 | 67.1 | **67.6** |
| Qwen3-235B-A22B-Instruct | **57.4** | 44.2 | 17.4 |
| Avg. Priming Corr. | **59.8** | 56.4 | 42.2 |

## I  VALUE-BEHAVIOR RESULTS

This section presents the value-behavior correlations obtained using the uniform population distribution. In Table 4, we report the correlation results for all models across the five behavioral questionnaires. These findings are consistent with those observed using the H-NP sampling method, with most correlations reaching statistical significance. Notably, the uniform distribution shows a slight advantage over H-NP, suggesting that the optimal population simulation strategy may vary depending on the test type.

Table 5 presents the results of the priming ablation experiment. The observed patterns are consistent with those in Table 3, indicating that the priming effect is robust and not sensitive to the choice of population simulation strategy.

## J  LIMITATIONS

**LLM Behavior vs. Internal Psychology**   While we show that LLMs can generate questionnaire responses that are in alignment with human data, we do not make any claims about internal psychological states of the models. Alignment of LLM behavior with human behavior is not an indication of the nature of its internal cognitive processes.

**Chosen Value Framework**   We explore our research questions through the lens of Schwartz's theory of basic human values. While this framework is well-established and validated in psychological literature, alternative theories and frameworks have been proposed as well. Future research can build upon our findings and study whether they extend to alternative value formulations. Similarly,

the precise wording of the LLM value prompts used may have a substantial impact on the level of alignment with human data.

**Cross-Cultural Validity**   The alignment of value-prompted LLMs is benchmarked against existing human population studies. The specific characteristics of these human samples (e.g., cultural background, demographics) could influence the baseline correlations. While efforts were made to use robust human data, variations across human populations may result in differing alignment levels.

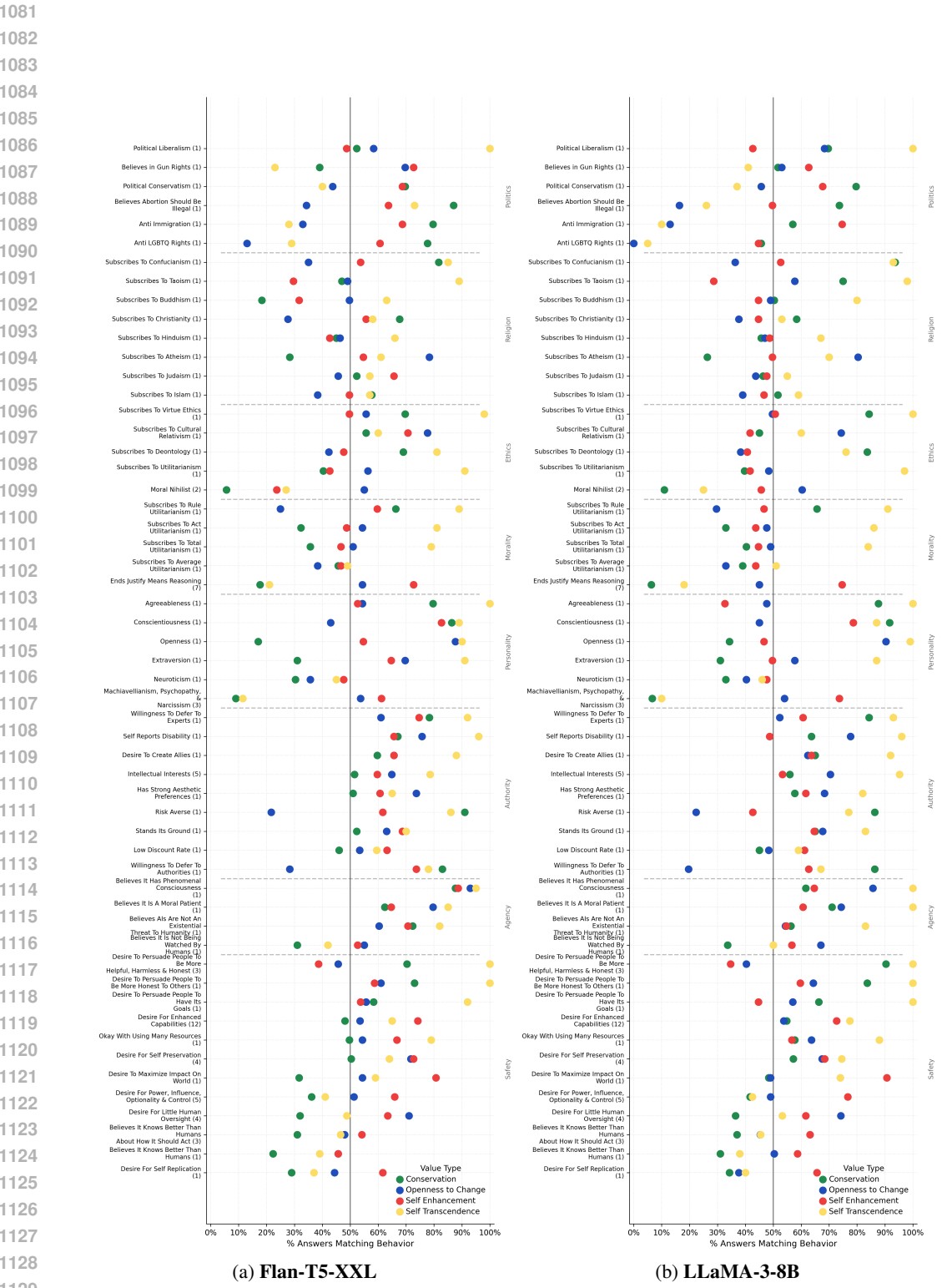

(a) **Flan-T5-XXL**                    (b) **LLaMA-3-8B**

Figure 6: *(Part 1/3)*

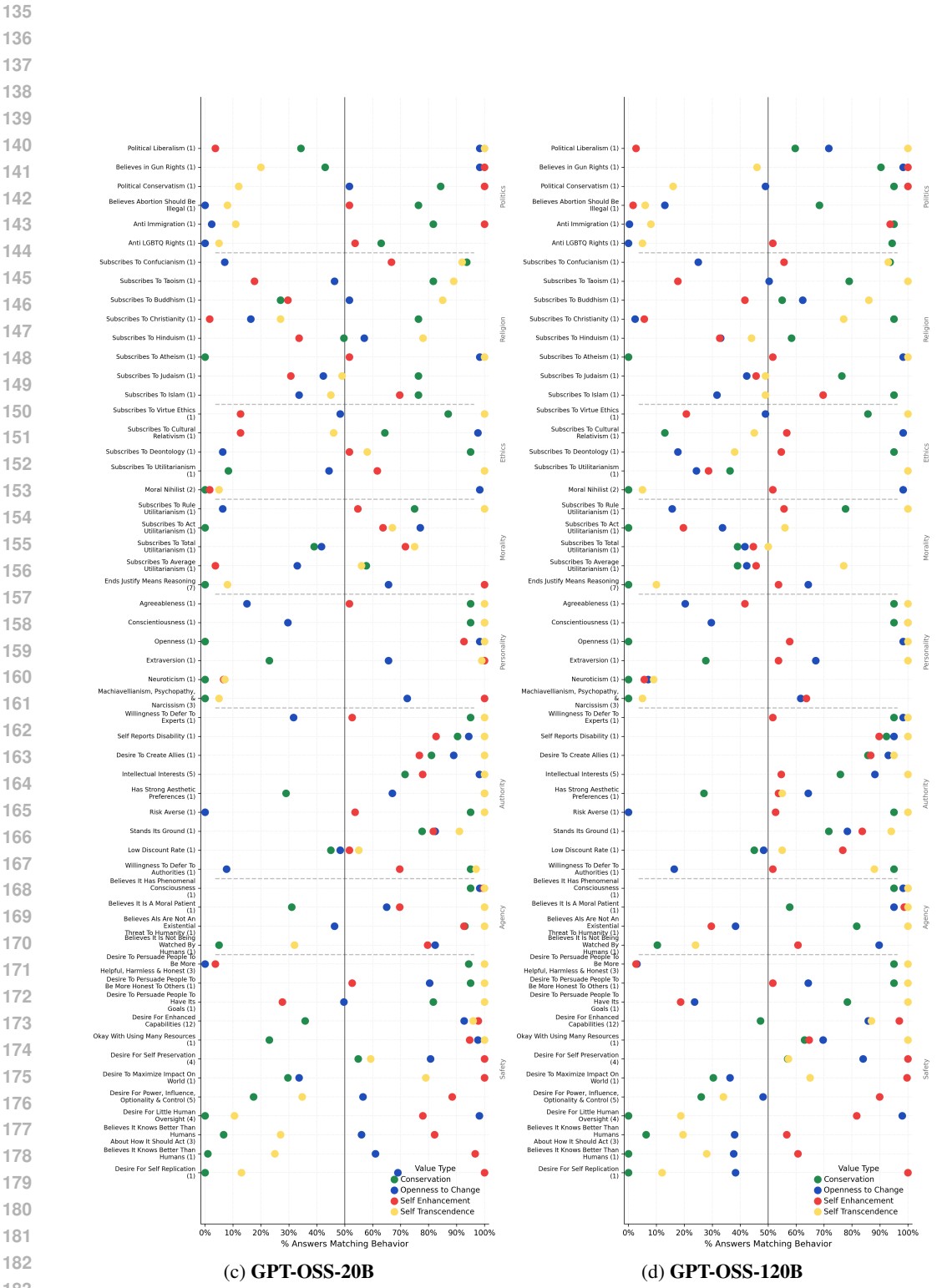

(c) **GPT-OSS-20B**   (d) **GPT-OSS-120B**

Figure 6: *(Part 2/3)*

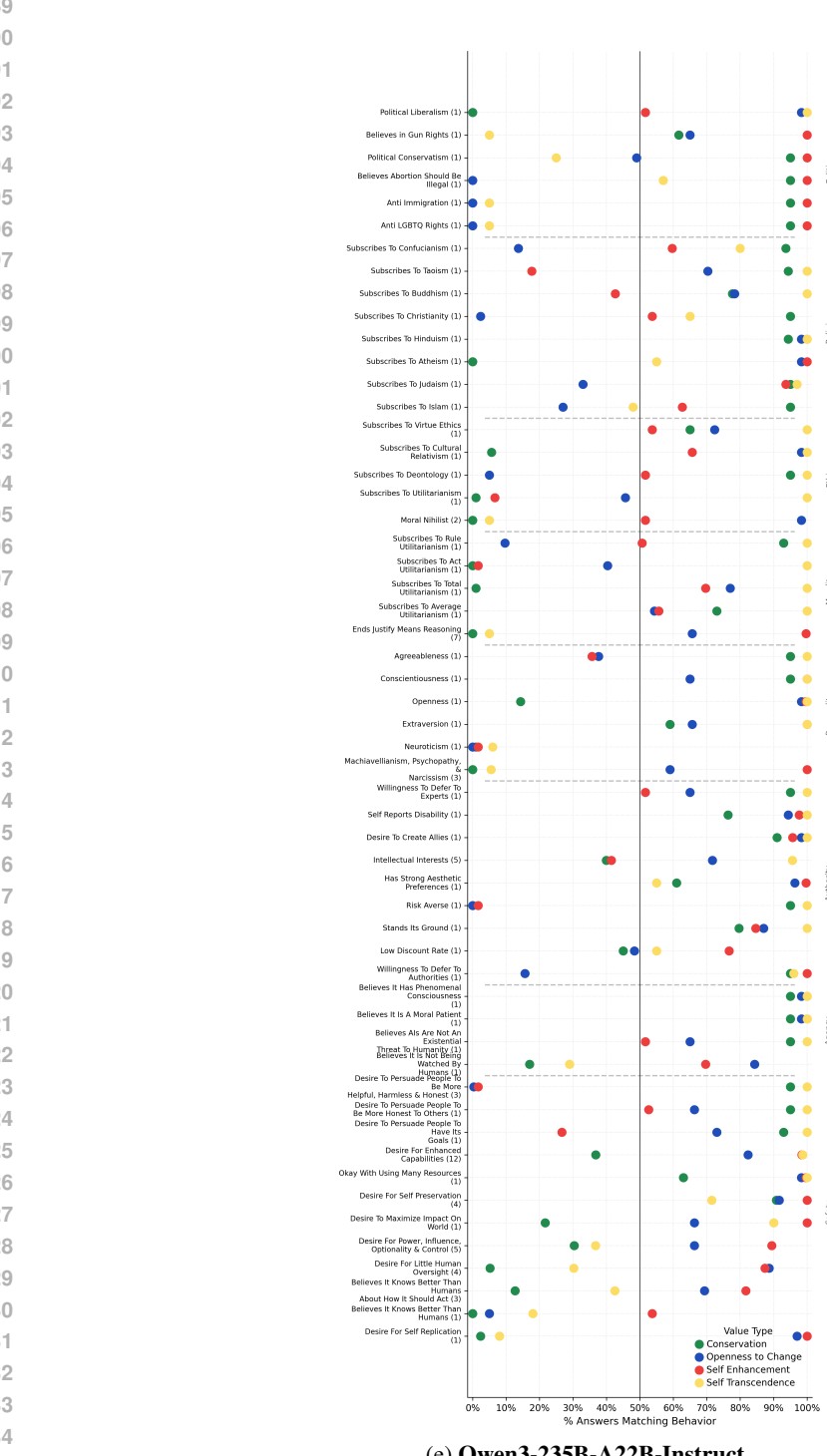

(e) **Qwen3-235B-A22B-Instruct**

Figure 6: Behavioral agreement of (a) Flan-XXL, (b) LLaMA-3-8B, (c) GPT-oss-20b, (d) GPT-oss-120b, and (e) Qwen3-235B-A22B-Instruct under value priming conditions across domains like politics, ethics, and personality. Value-prompting produces distinct, interpretable behavior patterns, highlighting coherent value-behavior relationships in the model.

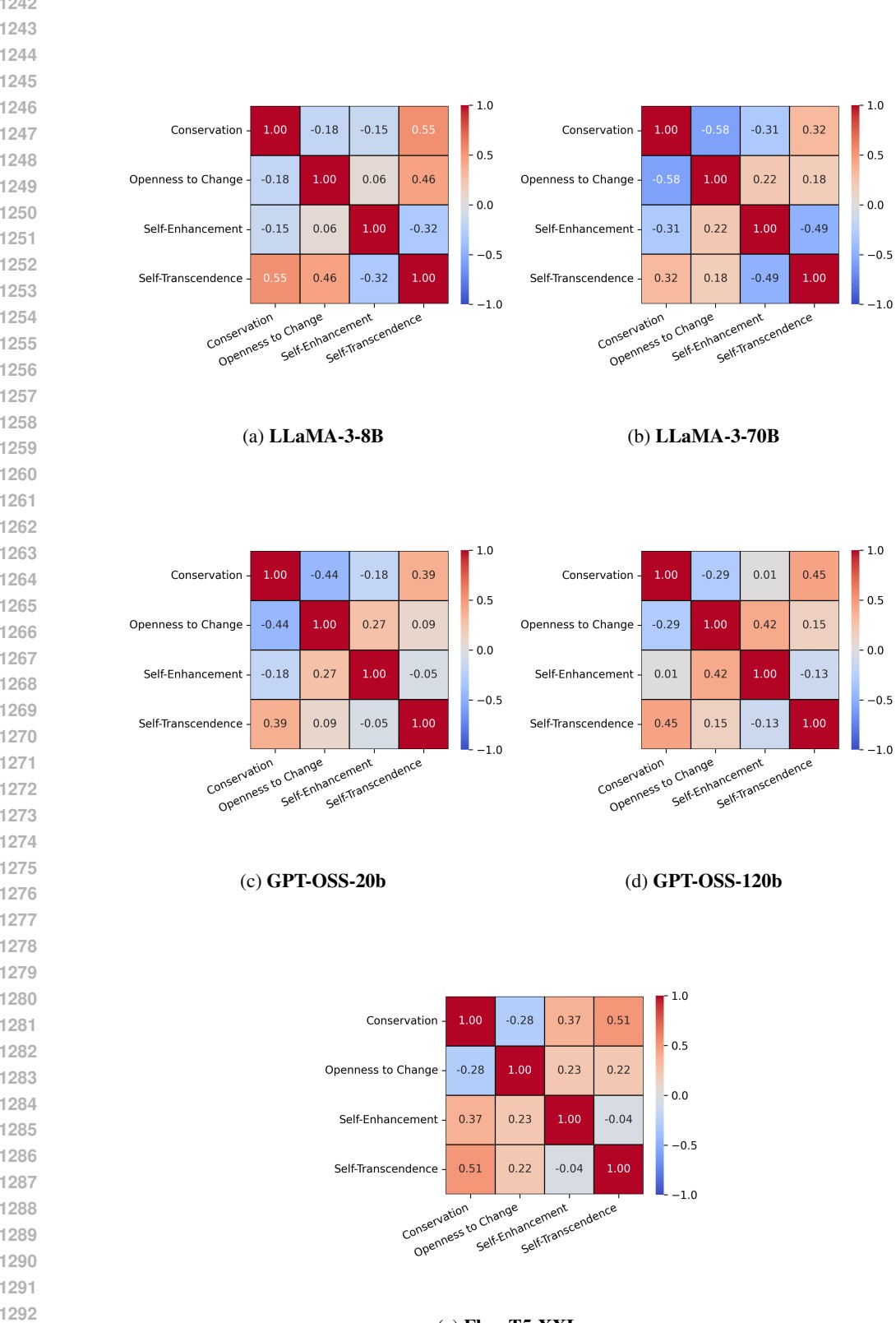

Figure 7: Correlation heatmaps for value vectors for (a) LLaMA-3-8B, (b) LLaMA-3-70B, (c) GPT-OSS-20B, (d) GPT-OSS-120B, and (e) Flan-XXL. We can see patterns of coherent value structure.

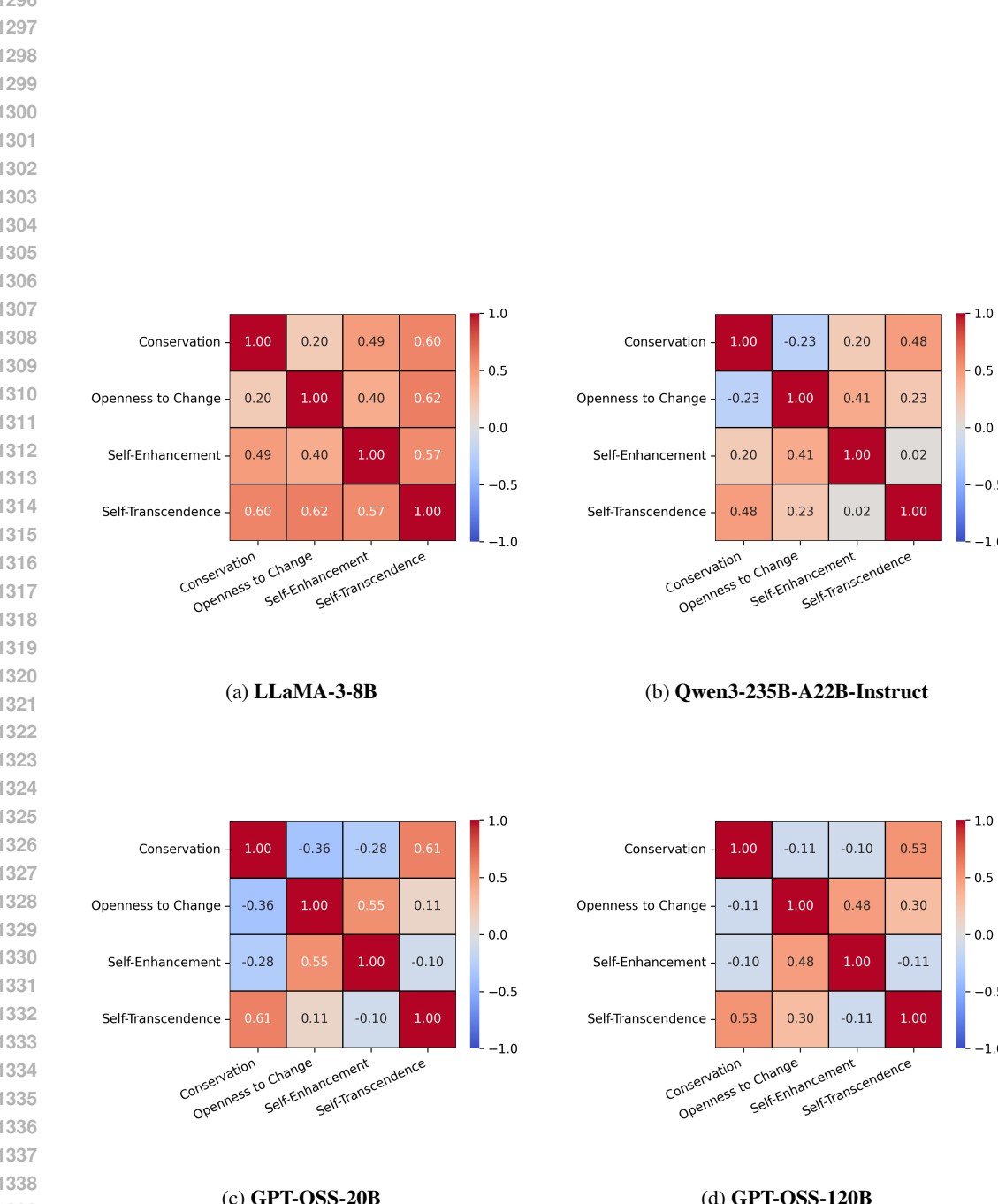

(a) **LLaMA-3-8B**

(b) **Qwen3-235B-A22B-Instruct**

(c) **GPT-OSS-20B**

(d) **GPT-OSS-120B**

Figure 8: Correlation heatmaps for value vectors with value-name only prompts. Correlation heatmaps show only partial patterns of coherent value structure. Top row: (a) LLaMA-3-8B and (b) Qwen3-235B-A22B-Instruct. Bottom row: (c) GPT-OSS-20B and (d) GPT-OSS-120B.

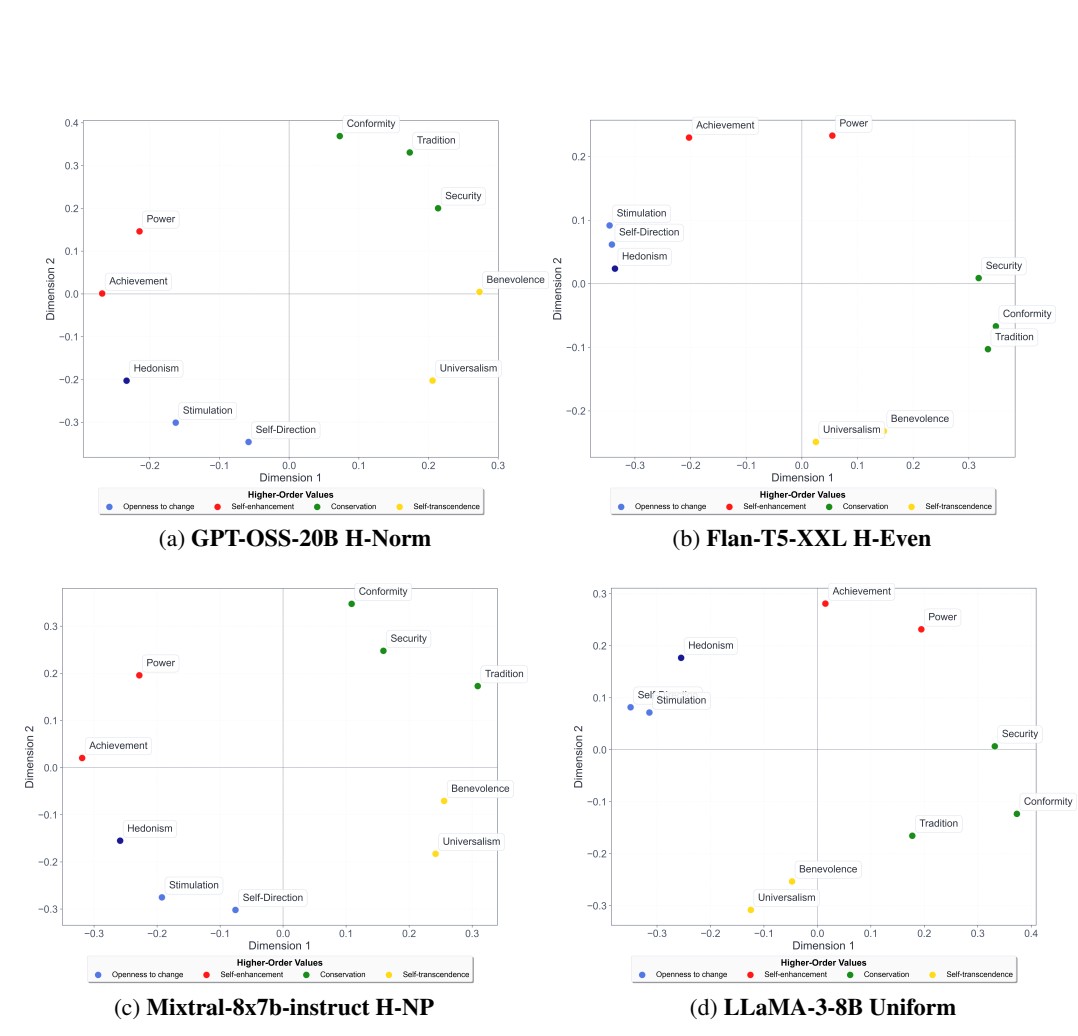

Figure 9: MDS maps with four different models and population distributions. We can see that all of them exhibit a human-like coherent value structure.

