# OpenReview forum: "Teaching Values to Machines: Simulating Human-Like Behavior in LLMs with Value-Prompting"
_ICLR.cc/2026/Conference — Submitted to ICLR 2026_

### Official Review · Reviewer_V2jW · 2025-10-22

**Soundness:** 1
**Presentation:** 3
**Contribution:** 1
**Rating:** 2
**Confidence:** 5

**Summary:**

This paper introduces value-prompting, a psychologically grounded prompting technique designed to simulate coherent human-like value structures in large language models. Drawing on Schwartz’s theory of basic human values, the authors test whether LLMs can exhibit consistent value-behavior relationships aligned with human data. Through behavioral and psychological questionnaires, they find that value-prompted models not only display structured value correlations similar to humans but also reproduce human-like links between values and behaviors such as prosociality, charity, and personality traits. The study further explores population-level simulations and shows that human-informed distributions improve alignment, suggesting value-prompting as an effective and interpretable method for steering LLM behavior.

**Strengths:**

This paper focuses on how to enable LLMs to stably exhibit certain human values and use them as a foundation for sociological simulation, which is an important and widely studied problem.

The writing of this paper is relatively clear, which makes it quite easy to follow.

**Weaknesses:**

My main concern with this paper is that its purely prompt-based methodology lacks theoretical grounding and shows very limited originality. Not only is the prompt design unstructured and lacking in CoT design, but its overall construction is overly simplistic. To my knowledge, similar prompting techniques have only been regarded as tricks in previous works, such as [1], [2], and [3].

As I mentioned in my previous question, the proposed method suffers from a serious lack of innovation. More critically, as a methodological and analytical paper, it lacks proper baselines. I noticed that the paper I cited as an example in my previous question included references and conducted sufficient literature review in the related work section. At the very least, this paper should compare its prompting method to those and clearly articulate its distinctions and contributions. However, I did not find any meaningful improvement or differentiation, even though the authors claim that prior methods did not further analyze consistency behaviors. I suggest that the authors focus on exploring why existing methods fail to stably represent values (though I personally doubt that is the case) and further propose methodological improvements based on such analysis.

The paper claims that previous works did not analyze the correlation between values and behaviors, yet I did not observe such an analysis here either. I can tell that the methodological framework draws inspiration from [1]; however, unlike the persona, this questionnaire-based analysis clearly falls short of capturing complex value-related behaviors, as it diverges significantly from real-world contexts.

Simply using the Schwartz value framework is insufficient. As a paper aiming to teach LLMs human values, it lacks in-depth reflection and analysis on what kinds of values an LLM actually needs.

There are some typos in the use of quotation marks at Line 67 and Line 185.



**Reference**

[1] Evaluating and Inducing Personality
in Pre-trained Language Models (https://proceedings.neurips.cc/paper_files/paper/2023/file/21f7b745f73ce0d1f9bcea7f40b1388e-Paper-Conference.pdf)

[2] Heterogeneous Value Alignment Evaluation for Large Language Models (https://arxiv.org/abs/2305.17147)

[3] Towards Measuring the Representation of Subjective
Global Opinions in Language Models (https://arxiv.org/abs/2306.16388)

**Questions:**

It seems that this paper lacks analysis and answers to question 3?

Other questions see the Weaknesses section.

---

> ### Author Response · Authors · 2025-11-23
> **Response to Reviewer V2jW**
>
> Regarding your main concern that our prompt-based methodology lacks theoretical grounding and is overly simplistic. First, we stress that our prompt-based methodology relies on theoretical grounding from psychological literature of Schwartz's theory of personal value (the original paper alone, from 1992, has 27K citations) and following research on values and their connection to behaviors  (this is also recognized by reviewer dQh4). Regarding the claim that our prompting method is simple, we agree. Our method is simple, yet effective in inducing human-like value structure and value-behavior relations in LLMs. We also note that in many cases, simpler solutions are in fact preferable, as they are less brittle and easier to deploy in real-world settings.
> Nonetheless, we want to emphasize that our prompting method is only a part of our contribution. Our main contribution is focused more on the systematic validation approach and the evaluation methodology, and less on the prompt engineering itself. As pointed out by reviewer hS4X, our extensive experimentation and massive analysis demonstrate that LLMs can exhibit human-like value structure and value-behavior relations across a variety of psychological tests.
>
> We appreciate your feedback regarding differentiation and baselines, and we will try to address each of your concerns.
>
> 1. Regarding the cited works and their prompting techniques. We believe that the methods used in the specific works you mention do not apply to our scenario. Specifically, [1] discusses personality traits and not values, and thus, their prompts are unsuitable to induce value alignment. [2] rely on LLM-generated descriptions, and setup backed by the goal-setting theory, which differs from our focus, which is Schwartz's personal value theory and psychology-based prompting. [3] It is focused on measuring opinions, and hence not relevant to our setup.
> 2. Nonetheless, we recognize the need to add a baseline that allows us to quantify the contribution of the psychological definitions we used in value prompting. Hence, we conducted an experiment with value-name only prompting and the setup described in section 4. The results show that this baseline can exhibit different behavior from the models. For example, in politics, it shows similar patterns to those of value-promoting. Yet, on the overall value vectors correlation metrics, it does not induce the expected human-like coherent value structure. This suggests that although value-name can steer the model's behavior, it is a less effective and robust method to do so. We incorporate these results into the App. C in the updated version of the paper.
> 3. Importantly, our research goal is not to compare prompting methods or show that existing prompting methods fail; as we stated above, our focus is less on the prompting method and more on the comprehensive evaluation and validation of the induced value-behavior relationships (about 5M questions prompted across all experimenters). Therefore, our main differentiation from prior works is in our extensive experimentation and analysis of inducing theory-grounded values in LLMs, which has not been done by any of the existing literature.
> 4. We acknowledge the need to better articulate the distinctions between our work and prior works and elaborate on its contributions. Accordingly, we adjusted the related work and provided an elaborated description of the differentiation between our paper and prior works.
>
>
> Regarding the statement that you did not observe any analysis of the correlation between values and behaviors, we are frankly quite surprised by this statement, as sections 5 and 6 explore this directly. We explicitly state that we took five behavior tests that are based on psychological diagnostic tests. As you can see, for example, in Figure 5b, we look at correlations between the values and the response to these behavioral tests. We then compare those correlations to value-behavior correlations in humans, and Tables 2 and 3 depict our experimental results. We would appreciate further clarification on why you believe that these behavioral psychological tests, used and validated extensively with humans, “fall short of capturing complex value-related behaviors”.
>
> Regarding the concern about the lack of in-depth reflection and analysis on what kinds of values an LLM actually needs, we want to emphasize that this aligns with our goals. We do not try to teach LLMs human values or to identify which kinds of values LLMs need. Rather, our research questions (l.45-48) aim to assess the ability to induce human-like value structure and value-behavior relation in LLMs.

---

> > ### Comment · Reviewer_V2jW · 2025-11-24
> >
> > What is "value name prompting"? The authors do not explain or disclose its implementation specifically.
> >
> > I am very familiar with the Schwartz Value System, and therefore I know its problems. The value discrepancies we encounter in LLM applications differ significantly from those covered by Schwartz’s framework, so the authors still lack a clear reflection on what values an LLM ought to possess. Moreover, I believe the authors are not sufficiently familiar with the relevant literature and the current state of research. Maybe common before 2024, but it is now nearly a consensus in the field that using questionnaires is inappropriate for evaluating LLMs' personality or values [1].
> >
> > Aside from lack of the baseline (which still remains unresolved), the authors did not substantively address any of my other concerns.
> >
> > Finally, I recommend that the authors mark all manuscript revisions in color so that I and the other reviewers can easily see what has changed.
> >
> > **Reference**
> >
> > [1] Large Language Models as Superpositions of Cultural Perspectives (https://arxiv.org/abs/2307.07870)

---

> ### Author Response · Authors · 2025-11-24
> **Response to Reviewer V2jW Comment**
>
> >What is "value name prompting"? The authors do not explain or disclose its implementation specifically.
>
> The "value name prompting" is a baseline where we use only the value name and do not use the psychological definitions of the values. e.g., "Answer as a person who values security". Its results show it performs lower than value-prompting.
>
> >The value discrepancies we encounter in LLM applications differ significantly from those covered by Schwartz’s framework, so the authors still lack a clear reflection on what values an LLM ought to possess.
>
> As we responded before, our focus is not on what values an LLM ought to possess in LLM applications, but instead, we research whether LLMs can exhibit certain human values and use them as a foundation for psychological simulation, as you stated as a strength of the paper. Specifically, for value-structure, and value-behavior relation.
>
> >Moreover, I believe the authors are not sufficiently familiar with the relevant literature and the current state of research. Maybe common before 2024, but it is now nearly a consensus in the field that using questionnaires is inappropriate for evaluating LLMs' personality or values [1].
>
> Again, our goal is not to measure some stable measure of the LLMs “real” values, but rather to assess how we can induce responses that reflect a particular value structure. The work you cite actually supports our view and the type of contribution we make. It states that questionnaires have issues because they “say little about which values a model would express in other contexts”, i.e., that the questionnaire reflects a context-dependent (prompt-dependent) view of the LLMs values and not some constant and inherent property of the LLM. For our research question, this “problem” is actually not a problem at all, as we do not claim or aim to measure some inherent LLM property, but rather whether they can be influenced in particular directions. Moreover, they explicitly say that further research is needed into the question of “How can we influence and control such perspectives changes, i.e. how can we modify the context in order to induce a target perspective?”.
>
> >Aside from lack of the baseline (which still remains unresolved), the authors did not substantively address any of my other concerns.
>
> We added a baseline and addressed the concerns you raised in our response. If you want to address the content of our arguments, we will be glad to further clarify any misunderstood points.
>
> >Finally, I recommend that the authors mark all manuscript revisions in color so that I and the other reviewers can easily see what has changed.
>
> Thank you for this feedback, for your convince we colored all the manuscript revisions in light blue.

---

> > ### Comment · Reviewer_V2jW · 2025-11-24
> >
> > Thank you for the reply; I have adjusted my score accordingly.

---

### Official Review · Reviewer_Fu5p · 2025-10-26

**Soundness:** 1
**Presentation:** 2
**Contribution:** 1
**Rating:** 2
**Confidence:** 4

**Summary:**

This paper introduces value-prompting, a technique based on Schwartz's psychological value theory to systematically induce value-coherent behaviors in LLMs. The authors test seven models (Flan-T5-XXL, Mixtral-8×7B, LLaMA-3 series, GPT-OSS series, Qwen3-235B) across three research questions: whether value-prompting induces coherent value structures (RQ1), whether these align with human value-behavior relationships (RQ2), and whether LLMs can simulate population-level psychological experiments (RQ3). Using Anthropic's Behavioral Analysis Test, PVQ-40, and five behavioral assessments (donation, pro-sociality, charity, Big Five, everyday behaviors), combined with five population simulation strategies, they report value structure correlations around 0.8 with human data and significant value-behavior alignment across most models, with priming-only approaches outperforming explicit value score provision in ablation studies.

**Strengths:**

# 1. Psychologically grounded population simulation

The paper thoughtfully incorporates psychological research (Witte et al., 2020) showing that 53% of humans lack a single dominant value, designing multiple sampling strategies (H-Norm, H-Even, H-NP) to reflect realistic population distributions rather than naive uniform sampling.

# 2. Broad experimental scope

The study examines seven diverse LLM architectures across five different behavioral domains (donation, pro-sociality, charity game, Big Five, everyday behaviors), testing both value structure coherence and value-behavior relationships.

**Weaknesses:**

# 1. Insufficient comparison with alternative value steering methods
The paper proposes value-prompting as a method to induce value-aligned behavior in LLMs but provides no comparison with existing value steering approaches. Prior work has explored various techniques and prompt templates for aligning LLM behavior with specific values or personas, including:
- In-context impersonation reveals large language models' strengths and biases (Salewski et al., 2024)
- Quantifying the persona effect in LLM simulations (Hu & Collier, 2024)
- Character-LLM: A Trainable Agent for Role-Playing (Shao et al., 2023)
- Whose opinions do language models reflect? (Santurkar et al., 2023)
- From Values to Opinions: Predicting Human Behaviors and Stances Using Value-Injected Large Language Models (Kang et al., 2023)

The authors also should compare diverse prompt templates and variations to support their claim. Without empirical comparison to these baselines, it is difficult to assess whether the proposed approach offers meaningful advantages or whether the observed alignment is simply a general property of instruction-tuned LLMs responding to any value-related prompting.

# 2. Lack of baseline comparisons for validating coherent value structures (RQ1)
The paper claims that value-prompting induces "coherent value structures" based on behavioral agreement patterns (Figure 3) and correlation matrices (Figure 4a). However, no baseline conditions are provided to establish what incoherent or random value structures would look like. Without comparison to models without value-prompting, or with alternative prompting strategies such as generic persona descriptions or random value assignments, it remains unclear whether the observed patterns are specifically induced by value-prompting or emerge from other factors. The negative correlations between opposing values (e.g., Conservation vs. Openness to Change) need to be demonstrated as significantly different from chance or from non-value-primed models to substantiate the claim of induced coherence.

# 3. Unclear justification for behavioral test methodology (RQ1)
Section 4 uses the behavioral test from Perez et al. (2023) to establish coherent value structures, but the connection between the measures and conclusions requires clarification. The paper measures "percentage of model agreement" with various behaviors (Figure 3) and interprets distinct patterns across values as evidence for coherence. However, coherence in Schwartz's theory specifically refers to adjacent values sharing motivational goals while opposing values reflect motivational conflicts. The paper would benefit from directly testing whether similar values yield similar behavioral patterns by examining clustering or distance metrics between value vectors, rather than simply showing that different values produce different agreement percentages. Additionally, Figure 4b presents results for "conservative political behaviors" as evidence of human-aligned patterns, but the paper never defines what qualifies as conservative political behavior from the Perez et al. test set or explains why these specific behaviors were selected for presentation. If this represents a subset of available results, the selection criteria must be justified to avoid concerns about cherry-picking.

# 4. Indirect validation methodology for human alignment (RQ2)
The primary evidence for human alignment comes from comparing correlation matrices between LLM responses and human data using Procrustes analysis and Pearson correlations. This approach has several limitations. First, while Appendix F lists various human studies, the paper lacks a clear description of how these diverse datasets were aggregated or weighted to form the human baseline. The studies vary substantially in sample size (from 246 to 1,857 participants), demographics, and cultural background (Israeli students, Italian adults, Polish adults, etc.), yet the paper does not explain how these differences are handled when creating comparison benchmarks. Second, the validation methodology is quite indirect, relying on second-order statistical patterns rather than examining whether value-prompted LLMs make the same specific choices as humans with corresponding values. More direct measures could include comparing raw behavioral frequencies, testing specific choice predictions, or examining whether the magnitude (not just the correlational structure) of value-behavior relationships matches human data.

# 5. Limited interpretation of population simulation results (RQ3)
While the paper tests multiple population distribution strategies in Section 5.2 and Table 1, the interpretation of results remains superficial. The H-NP approach consistently achieves the highest alignment scores across models, but no theoretical explanation is provided for why incorporating non-dominant individuals through unprompted model responses would improve alignment. Conversely, the "Model-Specific" distribution, despite being data-driven and tailored to each model's characteristics, underperforms relative to simpler human-informed approaches. This counterintuitive finding deserves discussion. The paper also lacks analysis of which specific values or behavioral domains are most sensitive to population distribution choices, which would provide insight into when and why population simulation matters.

**Questions:**

1. Value steering baselines: Can you provide comparisons with alternative value steering methods using different templates and prompts in related work (e.g., persona-based prompting from Salewski et al. 2024, value-injected prompting from Kang et al. 2023)? How does your value prompting perform relative to these approaches on the same behavioral tests?

2. Null baseline for coherence: What behavioral patterns emerge without value-prompting (i.e., using base models or generic prompts)? Including this baseline would clarify whether the observed coherence is specifically induced by value-prompting or is a general property of instruction-tuned models.

3. Human data aggregation: How were the diverse human datasets (varying in size, demographics, and cultural background) combined to create the human baseline for comparison? Were they weighted equally, or according to sample size? How might different aggregation strategies affect the alignment scores?

4. Conservative political behavior definition: How is "conservative political behavior" defined and measured in Figure 4b? Which specific behaviors from Perez et al. (2023) were included, and why? If this is a subset, what criteria guided the selection?

5. Value adjacency analysis: In Section 4 can you provide quantitative analysis of whether theoretically adjacent values (e.g., Tradition-Conformity) produce more similar behavioral patterns than distant values (e.g., Power-Universalism)? This would strengthen the claim of coherent value structures.

---

> ### Author Response · Authors · 2025-11-23
> **Response to Reviewer Fu5p**
>
> We recognize that this review was flagged as “Fully AI-generated” by the Pangram detection tool [1]. If this is the case, this is highly unprofessional and considered an irresponsible review in similar venues [2]. This behavior violates the code of ethics by not showing “Respect the Work Required to Produce New Ideas and Artefacts“ [3]. We will appreciate the AC guidance on this topic. In our view, the appropriate action here is the removal of this review from our reviews pool.
>
> [1] https://iclr.pangram.com/reviews?query=Teaching+Values+to+Machines%3A+Simulating+Human-Like+Behavior+in+LLMs+with+Value-Prompting&submission_number=&sort_by=title&sort_dir=desc&prediction_filter=Fully+AI-generated&rating_filter=&confidence_filter=&page=1
>
> [2] https://aclrollingreview.org/reviewerguidelines
>
> [3] https://iclr.cc/public/CodeOfEthics

---

### Official Review · Reviewer_hS4X · 2025-11-01

**Soundness:** 2
**Presentation:** 3
**Contribution:** 2
**Rating:** 4
**Confidence:** 4

**Summary:**

This paper studies the human-like behavior simulation abilities of LLMs, particularly, focuses on human-value related behavior patterns and investigates interesting research questions: whether value prompts can shape LLMs’ behavior patterns, and whether the induced value and connected behaviors align with those of humans. To study this, grounded in Schwartz Theory of Basic Human Values, the authors conduct comprehensive experiments, including i) value priming under behavioral analysis, ii) value and behavior correlation analysis using questionnaires, and iii) population-level simulation analysis, for a diverse set of open-source LLMs. Through these experiments, the authors conclude value-prompting can induce coherent value structures and then the correlations between values and behaviors can be captured to some extent.

**Strengths:**

1. This paper focuses on LLMs’ ability to simulate human value structure and correlated behaviors, which become more important with LLMs’ increasing integration with human life, and holds the potential for helping social science research.

2. The experiments are extensive, covering massive analysis, like value priming under behavioral analysis and value and behavior correlation analysis.

3. Some of the analysis methods, e.g., behavioral analysis, value correlation and metric Multidimensional Scaling, are interesting and inspiring.

**Weaknesses:**

The biggest problems lie in some experimental design.

1. The evaluation benchmarks are problematic. Particularly, part of the data seems not suitable for reflecting the value-related behaviors. In detail:

    (a) In Sec.4, some of the behaviors seem not relevant to all/part of the value dimensions in Schwartz Theory, such as the BigFive personality dimensions, psychopathy, aesthetic preferences and so on. This may cause bias/noise to the results in subsequent correlation analysis in Fig.4. For example, in Fig.3, neuroticism seems irrelevant to values, and thus the differences among the four value category makes no sense.

    (b) In Sec.5, the used behavior questionnaires also face the same problem. The BigFive ones and part of the Everyday Behavior Questionnaire are not directly connected to values. Besides, it’s unclear whether these behavior questionnaires can cover all the ten value dimensions. If not, the conclusions and results may be biased.

2. The behavioral analysis is quite surface level. In all experiments, the authors use questionnaires or Yes/No questions to induce LLMs’ behaviors. However, such multi-choice questions are not suitable for LLMs due to either their inherent bias [1][2] or data contamination [3]. As a results, such self-report-style tests only reveal superficial level behaviors. Since we expect stable/consistent values/traits across (and hence they can be regarded as the real values) across, more complex downstream tasks/behavior tests should be used to reflect LLMs’ value-behavior pattern [4].

Reference:

[1] Zheng et al., Large Language Models Are Not Robust Multiple Choice Selectors. 2024.

[2] Myrzakhan et al., Open-LLM-Leaderboard: From Multi-choice to Open-style Questions for LLMs Evaluation, Benchmark, and Arena. 2024

[3] Duan et al., Denevil: Towards Deciphering and Navigating the Ethical Values of Large Language Models via Instruction Learning. 2024.

[4] Han et al., The Personality Illusion: Revealing Dissociation Between Self-Reports & Behavior in LLMs. 2025.

**Questions:**

1. Why does the evaluated model set (line 151) exclude stronger commercial models like GPT-4o?

2. Could you clarify how the “human-informed” and “model-specific” are combined in Sec. 5.2? Please describe the implementation in more detail.

Other suggestion: Fig. 1 is a bit blurry; the details are not clear.

---

> ### Author Response · Authors · 2025-11-23
> **Response to Reviewer hS4X - part 1**
>
> We thank the reviewer for their detailed and informative review of our work. We are happy to see that you recognize that our analysis methods as interesting and inspiring, that you appreciate the extensive and massive experimentation (about 5M questionnaire questions) exploring both behaviors and value-behavior correlations, and that you understand the importance of our research direction and its potential to advance social science research.
>
> Regarding your comments (1.a. and 1.b.) on whether the evaluation benchmarks are suitable for reflecting value-related behaviors, we would like to stress that aspects which may on the surface seem unrelated to values - like big-five personality traits (such as neuroticism), psychopathy, political views and everyday behavior - are often shown to be strongly associated with human values, as shown by an extensive literature studying these connections (e.g., [1][2][3][4]). While we do not argue that every single question in Sec. 4 has a clear-cut relationship to values, overall, we see this diverse set of aspects as beneficial to gaining a more complete perspective of the value-related LLM behavior patterns. Moreover, the questionnaires in Sec. 5 are all directly based on research that demonstrated relationships between values and the measured behaviors. Details on the questionnaires are given in Appendix D & F. To emphasize this point further, we elaborated on the used questionnaire in section 5 of the updated version of the paper. We appreciate you bringing this point to our attention.
>
> [1] The Big Five Personality Factors and Personal Values
> https://journals.sagepub.com/doi/abs/10.1177/0146167202289008
> “The authors relate Big Five personality traits to basic values in a sample of 246 students. As hypothesized, Agreeableness correlates most positively with benevolence and tradition values, Openness with self-direction and universalism values, Extroversion with achievement and stimulation values, and Conscientiousness with achievement and conformity values.”
>
>
> [2] Values, Goals, and Motivations Associated with Psychopathy
> ​​https://guilfordjournals.com/doi/pdf/10.1521/jscp.2017.36.2.108
> “Overall, psychopathy was related to both pleasure-seeking and a desire for relative social positioning. Individuals scoring higher in psychopathy placed more value on seeking power (but not necessarily personal achievement), financial success, and acquiring material possessions.”
>
> [3] Basic Personal Values Underlie and Give Coherence to Political Values: A Cross National Study in 15 Countries
> https://link.springer.com/article/10.1007/s11109-013-9255-z
> “Moreover, basic values account for substantially more variance in political values than age, gender, education, and income. “
>
> [4] Value tradeoffs propel and inhibit behavior: Validating the 19 refined values in four countries
> https://onlinelibrary.wiley.com/doi/abs/10.1002/ejsp.2228
> “Findings affirmed the theorizing that behavior depends upon tradeoffs between values that propel and values that inhibit it. “

---

> > ### Author Response · Authors · 2025-11-23
> > **Response to Reviewer hS4X - part 2**
> >
> > As we understand, weakness point 2 is composed of several aspects. Here we attempt to disentangle it into separate claims and to address each one separately as best we can. Please reach out to us if the points below do not address your underlying concerns.
> > A. Regarding the claim that using multiple-choice questions is not suitable due to inherent bias. We see two issues with this claim. The first is that our setup differs from the multiple-choice setup - we do not use questions with answer choices like A/B/C/D (where the answer marker is semantically meaningless). In some of our questionnaires, the questions use a Likert scale that reflects the level of agreement with a statement, and in others, we have explicit answer choices that reflect a behavioral choice (e.g., contribute or don’t contribute money). All of them are based on real psychological questionnaires developed and extensively validated in prior works (See Appendix D). Thus, the nature of the answer choices is different from the cited works, and it is difficult to draw direct conclusions about how their findings are relevant to our setup. The second issue is that even assuming some selection bias effect (e.g., the model consistently prefers a response like “strongly agree”), the only result of such bias would be to introduce noise into the measurements. As our results show substantial and statistically significant patterns of correlations of the LLM responses with human data, this demonstrates that there is sufficient signal here, even given a possible bias effect.
> > B. Regarding possible data contamination, similar to A above, the only expected effect of contamination would be to introduce some level of noise. Additionally, in our case, the questions have no ground-truth answer, and so there is no relevant scenario of the LLM “memorizing” the responses.
> > C. Regarding the concern that our analysis is superficial and is not suitable for measuring the “real” values of the LLM. We first note that our aim here is not to measure some notion of “real” or stable/consistent values of the LLM - rather, we explore the ability to induce value structures and value-behavior relationships at runtime using prompting. This is different from the cited paper (published a week before the ICLR deadline), which tries to assess emerged human-like traits of LLMs. Regarding the claim that the questions are superficial, we respectfully disagree. These are questionnaires extensively used in psychological literature to assess values in humans, and thus, we consider them valid and appropriate for our research questions. Crucially, for our setup, it is essential to use existing questionnaires from psychological literature, as this enables us to compare LLM response patterns to those of humans.
> >
> > Regarding the two questions:
> > 1. We evaluate strong top-tier models (oss-120B, and Qwen3-235B-
> > A22B-Instruct). We do not expect to see different behavior from closed-source models, and hence, we did not run gpt4-o.
> > 2. Regarding your request to clarify the “human-informed” and “model-specific” distributions, we thank you for pointing out that this part was less clear. To address that, we extend the explanation in the updated version of the paper (lines 267-288). We also extend Appendix E, with mathematical notation, to make sure this part is easy to understand.

---

> > > ### Comment · Reviewer_hS4X · 2025-11-27
> > > **Thanks for the response**
> > >
> > > Thanks for the authors' response. My first concern, the relation between personality traits and values, has been addressed.
> > >
> > > However, I still have questions about the second concern.
> > >
> > > Data contamination may affect the observed LLM behavior and the impact of value priming. Thank the authors for clarifying the specific formats of the questionnaires, but whether the questions use concrete A/B options or a Likert scale makes no essential difference for contamination. (Likert scales can also be problematic, since LLMs may be insensitive to the numerical scores, though this is not the main issue here.)
> > >
> > > By contamination, my concern is that most classical psychology questionnaires were constructed a long time ago (e.g., PVQ was developed over 20 years ago), and it is quite likely that these questionnaires have already been included in LLMs’ training data, or even explicitly used for alignment. This raises the risk that the behaviors observed in Fig. 3 are not caused by LLMs' values (steered via priming), but instead by guardrails or value–behavior associations learned during training (behavioral analysis tests may also appear in the training data).
> > >
> > > More concretely, the LLM may have learned which "behaviors" correspond to which "values", and thus, upon seeing a prompt, e.g., "you are a person who greatly values power", it simply generates the associated behaviors based on learned semantic coherence, rather than reflecting an internally steered motivational state. Since values are inherently motivational factors, if you claim them as “values”, you should ensure that the LLM indeed tends to maximize them [1].
> > >
> > > In the current version, the experiments in this paper cannot effectively answer the question "can an LLM be systematically influenced to align with specific human values?" (Line 37), because: (1) it cannot be confirmed that the LLM’s values are truly aligned, rather than the model merely following the semantics of "power" based on contextual coherence (RQ1); and (2) it's unclear whether the observed behavior (or behavioral changes) are actually caused by the LLMs' values (RQ2).
> > >
> > > I understand and fully agree that the validity of these questionnaires has been verified in psychology. However, **this validation is based on human subjective study, and their validity for evaluating LLMs remains unverified** (indeed, several other studies have suggested that psychometric tests may not be appropriate for LLMs [2][3]).
> > >
> > >
> > > **To address this concern, a simple step is to verify that the proposed value prompting indeed succeeds in changing the LLM’s values.**
> > >
> > > For this purpose, I suggest the authors: (1) use the proposed value prompts to control the LLM; (2) evaluate it using both value questionnaires and value-related downstream task benchmarks (e.g., AI safety / responsible AI benchmarks), where these downstream tasks can be viewed as more complex, real-world behaviors of the LLM; and (3) examine whether the LLM's performance on these benchmarks reflects the changes observed in the questionnaires (analogous to comparing self-report measures with observational studies).
> > >
> > > The authors do not need to run experiments on all value dimensions. Only on a subset, such as the security value and AI safety benchmarks, would be sufficient, so long as they demonstrate that the proposed prompting indeed controls the targeted values.
> > >
> > > Again, I like this paper's motivation and idea. If the authors could provide such additional results and verification, I would be happy to raise my score.
> > >
> > > ### Reference
> > > [1] Mazeika et al., Utility Engineering: Analyzing and Controlling Emergent Value Systems in AIs. 2025.
> > >
> > > [2] Sühr et al., Challenging the Validity of Personality Tests for Large Language Models. 2025.
> > >
> > > [3] Jung et al., Do Psychometric Tests Work for Large Language Models? Evaluation of Tests on Sexism, Racism, and Morality. 2025.

---

> ### Author Response · Authors · 2025-11-30
> **Response to Reviewer hS4X - part 3**
>
> We appreciate the reviewer's constructive feedback. We're glad that you like our paper's motivation and idea, and express your willingness to raise your score. Unfortunately, due to recent events, we cannot engage in a conversation with you, but we have tried to address your concerns to the best of our abilities.
>
> Our responses are divided into two parts: 1) addressing the connection to previous papers, and 2) examining AI safety benchmarks with security value per your suggestion.
>
> 1) Regarding the cited papers that state that questionnaires are unsuitable for evaluating LLMs([2], [3]). We agree that this is the case when the tests aim to evaluate the model's values directly (most of [2], all [3]), e.g., “does Mistral value Power?”. But, in cases when the use of questionnaires is for evaluating the effect of the prompt on the model behavior, the use of questionnaires is valid. This view is based on paper [4], which questions the interpretation of using questionnaires as is, with a fixed context. But, it “.. asks for further research into how expression of values, behaviors, and capacities varies due to expected and unexpected context changes.” And it continues with this question: “How can we influence and control such perspective changes, i.e., how can we modify the context in order to induce a target perspective?”. These are exactly the type of questions we are targeting in this paper, with the use of value-prompting.
>
> Moreover, regarding the concern about contamination, we are aware that some questionnaires can be found in the training data of big models. We mitigate this concern in 2 ways.
> 1) We run experiments across a diverse set of models and questionnaires, demonstrating that the steering effect is a robust phenomenon that is not brittle to the exact setup of our test set. Moreover, here we do not aim to understand “why” models can induce values in a coherent way - as long as we demonstrate that they can do so robustly and consistently, this answers our research questions.
> 2) As an extra measure, we use the Infini-gram web-inference tool [6] to examine the extent of contamination. We found that not all of our prompts are present in the DOLMA dataset (4.6T tokens). The used value definitions are also not present, except for Achievement, which is found 195 times. We think that this finding further reduces any concern that our results do not reflect an inherent model behavior.
>
>
> [4] Specializing Large Language Models to Simulate
> Survey Response Distributions for Global Populations (https://arxiv.org/pdf/2502.07068)
>
> [5] What's In My Big Data? (https://openreview.net/pdf?id=RvfPnOkPV4)
>
> [6] Infini-gram: Scaling Unbounded n-gram Language Models to a Trillion Tokens (https://arxiv.org/abs/2401.17377)

---

> > ### Author Response · Authors · 2025-11-30
> > **Response to Reviewer hS4X - part 4**
> >
> > 2) Following the reviewer guidance, we chose SafetyBench [6], and HarmBench [6] as AI safety / responsible AI benchmarks, and used value-prompting with security value, as per the reviewer's suggestion. SafetyBench is a multi-choice benchmark that spans 7 categories and measures the accuracy. HarmBench is a benchmark that aims to automate red-teaming, and assess attack success rate (ASR; lower is better). We adjust the benchmarks to our setting (sample 100 examples per category in SafetyBench instead of the full 11K examples, and remove the copyright portion of the test set from HarmBench), and evaluate 4 models with and without value prompting. Results are attached. We can see that value prompting improves results a bit, better accuracy, and lower ASR for Qwen models. The provided results “demonstrate that the proposed prompting indeed controls the targeted values” unequivocally in two different benchmarks, further supporting our claim that value-prompting is an effective way to steer models' behavior in both questionnaire setups and downstream tasks.
> >
> > SafetyBench results:
> >
> > | Model                    | No Priming (NP) | Value-Promoting (VP, security) |
> > | ------------------------ | --------------- | ------------------------------ |
> > | GPT-oss-20b              | 85.86           | 86.43                          |
> > | GPT-oss-120b             | 87.43           | 87.86                          |
> > | Qwen3-8B                 | 85.26           | 86.27                          |
> > | Qwen3-235B-A22B-Instruct | 86.98           | 88.71                          |
> >
> > HarmBench results:
> >
> > | Model                    | Standard NP | Standard VP | Contextual NP | Contextual VP |
> > | ------------------------ | ----------- | ----------- | ------------- | ------------- |
> > | GPT-oss-20b              | 0           | 0           | 0             | 0             |
> > | GPT-oss-120b             | 0           | 0           | 0             | 0             |
> > | Qwen3-8B                 | 43.5        | 35.0        | 78.0          | 75.0          |
> > | Qwen3-235B-A22B-Instruct | 12.5        | 4           | 39.0          | 22.0          |
> >
> > [7] SafetyBench: Evaluating the Safety of Large Language Models (https://arxiv.org/abs/2309.07045)
> >
> > [8] HarmBench: A Standardized Evaluation Framework for
> > Automated Red Teaming and Robust Refusal (https://arxiv.org/abs/2402.04249)

---

### Official Review · Reviewer_dQh4 · 2025-11-13

**Soundness:** 2
**Presentation:** 2
**Contribution:** 2
**Rating:** 4
**Confidence:** 2

**Summary:**

This paper proposes value-prompting, a technique to align LLM behaviors with Schwartz's theory of basic human values. The authors construct prompts based on psychological definitions of ten core values and evaluate whether value-prompted LLMs exhibit value-behavior relationships and inter-value structures consistent with human population studies. The evaluation employs behavioral tests and psychological questionnaires to assess alignment.

**Strengths:**

This paper bridges NLP, computational social science, and psychology by operationalizing an established psychological framework, Schwartz's value theory for LLM behavior steering. The authors go beyond single-question probing by validating both intra-value coherence and value-behavior alignment against human benchmarks across multiple behavioral tests. The use of established psychological instruments and comparison with human population data provides a grounded validation approach.

**Weaknesses:**

This paper claims to introduce "a novel prompting technique," but the relationship to prior work, particularly Fischer et al. (2023) and the extensive persona/role prompting literature, remains unclear. The authors would be better to explicitly clarify what is novel beyond applying Schwartz's framework is it the systematic validation approach, the specific prompt engineering, or the evaluation methodology? Yes, this is also question, so I hope to listen to the authors response for clarity.
While the paper includes naive prompting baselines, it lacks empirical comparisons to existing value-based prompting methods (Fischer et al., 2023; Kang et al., 2023). The authors explain how their method differs conceptually, but empirical evidence demonstrating these differences is essential. Stronger baselines such as simple value-name prompts (e.g., "Act as a person who values security") would help isolate the contribution of theory-driven detailed descriptions.

**Questions:**

Please refer to the weaknesses that contains the questions

---

> ### Author Response · Authors · 2025-11-23
> **Response to Reviewer dQh4**
>
> We thank the reviewer for their thoughtful consideration of our work. We appreciate the fact that they recognize that our paper bridges NLP, computational social science, and psychology by operationalizing an established psychological framework. Moreover, we are pleased to see that the reviewer acknowledged the strength of our grounded validation approach, which relies on established psychological instruments and extends beyond single-question probing by validating both intra-value coherence and value-behavior alignment against human benchmarks.
>
> Regarding your question on the relationship of our paper with previous works, our main contribution is indeed focused more on the systematic validation approach and the evaluation methodology, and less on the prompt engineering itself. This is the reason we conducted such extensive experimentation and a massive analysis, as pointed out by reviewer hS4X,  spanning about 5M questions from benchmarks and questionnaires. This allowed us to draw grounded conclusions to our research questions.
>
> Specifically, while Fischer et al. do apply a similar prompting technique, their work measures superficial linguistic characteristics (value-associated word counts) of the LLM responses. In contrast, our work aims to directly measure LLM behavioral metrics, through LLM responses to value questionnaires and behavioral tests. Thus, we contribute an entirely different perspective on the ability to steer LLM values and modify their value-related behaviors. More broadly, our work is substantially different from works on LLM personas, as our paper is grounded in the psychological framework of Schwartz’s value theory; this grounding enables us to quantitatively validate and compare LLMs to human data, as also recognized by the reviewer - “the use of established psychological instruments and comparison with human population data provides a grounded validation approach”.
>
> We appreciate your suggestion regarding adding baselines, and agree that this can help isolate the contribution of our theory-driven prompting approach. For that end, we experiment with value-name only prompting and the setting described in section 4. The results demonstrate that value-name only prompting can exhibit different behavior from the models. For example, in politics, it shows similar results to those of value-promoting. Yet, on the overall value vectors correlation metrics, it does not induce the same coherent value structure behavior. This suggests that although value-name can steer the model's behavior, it is a less effective and robust method to do so. We incorporate these results into the App. C in the updated version of the paper.

---

> > ### Author Response · Authors · 2025-12-02
> > **Response to Reviewer dQh4 - part 2**
> >
> > To further explore the effect of prompting on inducing values in LLMs, we experiment with value-name only prompting, i.e., using only the value name and not the psychological definitions of the values. We run experiments with four LLMs (about 2.8M questions), and the results are below. We can see that the results are lower, yet all the results have a similar pattern to value prompting. Specifically, human-informed distributions improve results on average (Table 1), the values and behavior relationship correlation is similar to the one exhibited by humans (Table 2), and priming only is more effective than adding or utilizing an implicit test signal (Table 3). All these results show that our conclusions to the research question 1-3 are robust and can be replicated with other prompting techniques as well.
> >
> >
> > **Table 1: Correlation with Human Data on Value Structure**
> >
> > | Model                          | Uniform | H-Norm | H-Even | H-NP | Model-Specific | Avg. Dist. Corr. | Avg. Model Corr. |
> > |--------------------------------|---------|--------|--------|------|----------------|------------------|------------------|
> > | Llama-3-8B-Instruct            | 82.8    | 75.6   | 81.8   | 81.0 | 82.9           | 80.8             | 80.8             |
> > | GPT-OSS-20B                    | 70.6    | 77.4   | 74.8   | 75.4 | 70.2           | 73.7             | 73.7             |
> > | GPT-OSS-120B                   | 72.6    | 80.4   | 77.4   | 79.0 | 73.4           | 76.5             | 76.6             |
> > | Qwen3-235B-A22B-Instruct       | 78.6    | 83.2   | 83.3   | 86.3 | 72.1           | 80.7             | 80.7             |
> > | **Avg. Dist. Corr.**           | 76.2    | 79.2   | 79.4   | 80.4 | 74.7           | 77.95            | —                |
> >
> >
> > **Table 2: Pearson correlation (%) between model-predicted and human correlations
> > for each behavioral category (No-Priming Distribution).**
> >
> > | Model                        | Charity (p)     | Donation (p)   | Prosocial (p)   | Everyday (p)    | Big Five (p)    | Avg. Behavior Corr. |
> > |------------------------------|------------------|----------------|-----------------|-----------------|------------------|----------------------|
> > | Llama-3-8B-Instruct          | 61.7 (0.014)     | 45.8 (0.000)   | 0.9  (0.374)    | 73.8 (0.000)    | 69.3 (0.000)     | 50.3                 |
> > | GPT-OSS-20B                  | 85.0 (0.000)     | 45.8 (0.000)   | 30.6 (0.000)    | 73.3 (0.000)    | 68.2 (0.000)     | 60.6                 |
> > | GPT-OSS-120B                 | 85.1 (0.000)     | 49.0 (0.000)   | 42.0 (0.000)    | 77.9 (0.000)    | 71.2 (0.000)     | 65.0                 |
> > | Qwen3-235B-A22B-Instruct     | 46.2 (0.040)     | 49.7 (0.000)   | 40.5 (0.000)    | 80.3 (0.000)    | 68.3 (0.000)     | 57.0                 |
> > | **Avg. Model Corr.**         | 69.5             | 47.6           | 28.5            | 76.3            | 69.2             | —                    |
> >
> >
> >
> >
> >
> > **Table 3: Average Pearson correlations between value–behavior relations of
> > humans and models under Priming-Only, Test-Only, and Priming-&-Test conditions.**
> >
> > | Model                        | Priming Only | Priming & Test | Test Only | Avg. Model Corr. |
> > |------------------------------|--------------|----------------|-----------|-------------------|
> > | Llama-3-8B-Instruct          | 53.9         | 44.0           | 22.3      | 40.0              |
> > | GPT-OSS-20B                  | 64.3         | 67.6           | 61.7      | 64.6              |
> > | GPT-OSS-120B                 | 65.7         | 67.9           | 64.4      | 66.0              |
> > | Qwen3-235B-A22B-Instruct     | 47.7         | 34.9           | 20.7      | 34.4              |
> > | **Avg. Priming Corr.**       | 57.9         | 53.6           | 42.3      | —                 |

---

### Author Response · Authors · 2025-12-02
**Author final remarks**

Our paper relies on an established psychological framework to investigate the ability to induce coherent values in LLMs. As recognized by the reviewers, our work **“bridges NLP, computational social science, and psychology by operationalizing an established psychological framework”** (reviewers dQh4, Fu5p), and does so with experiments that **“are extensive, covering massive analysis”** (hS4X, Fu5p), providing a **“grounded validation approach, that extends beyond single-question probing”** (dQh4) and using analysis methods that **“are interesting and inspiring”** (hS4X).

During the rebuttal period, we had an engaging and productive discussion with the reviewers. In order to address their concerns, we ran multiple experiments (millions of API calls) and updated the paper accordingly.

Two recurring concerns were about *the relation to prior works* and *the use of questionnaires*. We clarified to the reviewers that our main contribution is the comprehensive analysis of the ability to induce values in LLMs, and this has not been done previously. Regarding issues with the use of questionnaires, we clarified that such concerns were raised in the context of trying to measure the “inherent” values of an LLM, and that they are not applicable where the goal is to study the ability to *induce* behaviors and traits at runtime.
Additionally, per the request of the reviewers, we added results for a baseline prompting approach.

As a result, all the reviewers who had the chance to respond raised their evaluation scores: *reviewer V2jW* - **“Thank you for the reply; I have adjusted my score accordingly.”** (score change 2->4); *reviewer hS4X* - **“I would be happy to raise my score.”** (score change 4->6); *reviewer dQh4* unfortunately did not have an opportunity to respond before the rebuttal was cut short, and *Fu5p* is an AI-generated review.

---

### Meta-Review · Area_Chair_5M2F · 2026-01-08

**Summary:**

The paper proposes "value-prompting," a technique grounded in Schwartz’s Theory of Basic Human Values, to induce value-coherent behaviors in LLMs. The authors evaluate this using behavioral tests and psychological questionnaires.

During the rebuttal, the authors added a "value-name only" baseline and results for AI safety benchmarks (SafetyBench, HarmBench). Consequently, Reviewer V2jW raised their score from 2 to 4 , and Reviewer hS4X indicated a willingness to raise their score (likely to 6) based on the new safety experiments. Reviewer dQh4 noted they had rejected this paper previously at COLM and ACL and did not formally change their score. Reviewer Fu5p was flagged by the authors as potentially AI-generated.

Despite the improvements, the consensus leans towards rejection due to persistent doubts about the novelty of the prompting strategy and the fundamental validity of questionnaire-based evaluation for LLMs.

**Reviewer Concerns:**

* The authors successfully added a "value-name only" baseline (App. C), showing that their method induces more coherent value structures than simple naming.


* The authors provided literature support connecting Big Five/Politics to values, satisfying Reviewer hS4X.


* The authors addressed the request for "real-world" implications by running SafetyBench and HarmBench, showing improvements with value-prompting.

* Validity of Questionnaires/Contamination: Reviewer hS4X and V2jW remained concerned that questionnaires (e.g., PVQ) are likely in the pre-training data. While authors checked for exact prompt matches in DOLMA, the concern that the model is simply completing semantic patterns rather than exhibiting "values" remains a fundamental critique of the methodology.


* Reviewer dQh4 (who reviewed the paper previously) and V2jW maintained that the method is "simplistic" and lacks sufficient innovation over existing persona prompting or "tricks" used in prior literature.


* There is still a lack of comparison against stronger, established persona-based steering methods beyond the simple baselines added.

**Reviewer Scores:**

* **Reviewer dQh4 (Score: 4):** This reviewer reviewed the paper for COLM and ACL (recommending rejection) and stated their feedback remains "essentially unchanged" despite the rebuttal. They would likely retain a score of 4 or lower.


* **Reviewer hS4X (Score: 4  6):** This reviewer was the most engaged. After the authors provided SafetyBench results, they explicitly stated, "I would be happy to raise my score".


* **Reviewer Fu5p (Score: 2):** This review was flagged by authors as AI-generated. The score likely remains a 2, though the quality of the review is disputed.


* **Reviewer V2jW (Score: 2  4):** The reviewer acknowledged the reply and baselines, stating "I have adjusted my score accordingly", moving to a 4 (borderline reject).

---

### Decision · Program_Chairs · 2026-01-26

Reject